# Distinct regions of *H. pylori*'s bactofilin CcmA regulate protein–protein interactions to control helical cell shape

**Sophie R Sichel[1,2], Benjamin P Bratton[3,4], Nina R Salama[1]\***

[1]Division of Human Biology, Fred Hutchinson Cancer Center, Seattle, United States; [2]Molecular Medicine and Mechanisms of Disease Graduate Program, University of Washington, Seattle, United States; [3]Department of Pathology, Microbiology and Immunology, Vanderbilt University Medical Center, Nashville, United States; [4]Vanderbilt Institute for Infection, Immunology and Inflammation, Nashville, United States

**Abstract** The helical shape of *Helicobacter pylori* cells promotes robust stomach colonization; however, how the helical shape of *H. pylori* cells is determined is unresolved. Previous work identified helical-cell-shape-promoting protein complexes containing a peptidoglycan-hydrolase (Csd1), a peptidoglycan precursor synthesis enzyme (MurF), a non-enzymatic homolog of Csd1 (Csd2), non-enzymatic transmembrane proteins (Csd5 and Csd7), and a bactofilin (CcmA). Bactofilins are highly conserved, spontaneously polymerizing cytoskeletal bacterial proteins. We sought to understand CcmA's function in generating the helical shape of *H. pylori* cells. Using CcmA deletion analysis, in vitro polymerization, and in vivo co-immunoprecipitation experiments, we identified that the bactofilin domain and N-terminal region of CcmA are required for helical cell shape and the bactofilin domain of CcmA is sufficient for polymerization and interactions with Csd5 and Csd7. We also found that CcmA's N-terminal region inhibits interaction with Csd7. Deleting the N-terminal region of CcmA increases CcmA-Csd7 interactions and destabilizes the peptidoglycan-hydrolase Csd1. Using super-resolution microscopy, we found that Csd5 recruits CcmA to the cell envelope and promotes CcmA enrichment at the major helical axis. Thus, CcmA helps organize cell-shape-determining proteins and peptidoglycan synthesis machinery to coordinate cell wall modification and synthesis, promoting the curvature required to build a helical cell.

**\*For correspondence:**
nsalama@fredhutch.org

**Competing interest:** The authors declare that no competing interests exist.

## Editor's evaluation

This study addresses an important unresolved question, the mechanisms governing cell shape in a helical organism. Particularly exciting is the observation that a bactofilin – typically viewed as a cytoskeletal protein – plays a key role in modulating the balance between peptidoglycan synthesis and degradation necessary to generate the helical shape. These findings enhance our understanding of cell wall synthesis in bacteria generally, particularly with regard to the importance of local phenomena for cell shape generation.

## Introduction

*Helicobacter pylori* is a helical-shaped, Gram-negative bacteria that infects more than half of the world's population (*Hooi et al., 2017*). Chronic infection with *H. pylori* can cause gastric ulcers and cancers; approximately 80% of gastric cancer cases are attributed to *H. pylori* infection (*de Martel et al., 2020*). The helical shape of *H. pylori* cells helps establish colonization of the stomach and

promotes immunopathology during infection (*Martínez et al., 2019*; *Montecucco and Rappuoli, 2001*; *Sycuro et al., 2010*; *Sycuro et al., 2012*; *Yang et al., 2019*). Multiple proteins have been identified that are essential for the helical shape of *H. pylori* cells, most of which influence the shape and composition of the peptidoglycan (PG) sacculus directly or indirectly (*Sycuro et al., 2010*; *Sycuro et al., 2013*; *Sycuro et al., 2012*; *Yang et al., 2019*).

Several cell-shape-determining (Csd) proteins form membrane-spanning protein complexes. Csd5 is a transmembrane protein that binds directly to PG in the periplasm with its SH3 domain and binds to CcmA in the cytoplasm. Csd5 also interacts with ATP synthase and MurF, an essential PG precursor enzyme that catalyzes the final cytoplasmic PG biosynthesis step, independent of interactions with CcmA (*Blair et al., 2018*; *Hrast et al., 2013*). Another complex is formed by Csd7, Csd1, and Csd2. Csd7 is a transmembrane protein that stabilizes the PG-hydrolase Csd1, and Csd2, a catalytically inactive homolog of Csd1. Weak interactions between Csd7 and CcmA have been detected by co-immunoprecipitation experiments that include cross-linking reagents followed by mass spectrometry (*Yang et al., 2019*), but whether CcmA, Csd7, and Csd5 are all part of one protein complex was unclear. CcmA and probes that report on PG synthesis localize to the major helical axis of *H. pylori* cells (regions with a Gaussian curvature value of 5 ± 1 μm$^{-2}$ on a helical cell), indicating that CcmA is in the same region of the cell where increased levels of PG synthesis occur (*Taylor et al., 2020*).

CcmA is the only bactofilin encoded in the *H. pylori* genome. Bactofilins are a class of highly conserved, spontaneously polymerizing, cytoskeletal proteins in bacteria that perform a variety of functions (*Bulyha et al., 2013*; *El Andari et al., 2015*; *Lin et al., 2017*; *Osorio-Valeriano et al., 2019*) including modulating cell shape in multiple different bacterial species (*Brockett et al., 2021*; *Hay et al., 1999*; *Jackson et al., 2018*; *Koch et al., 2011*; *Sycuro et al., 2010*) and morphogenesis of cell appendages (*Billini et al., 2019*; *Caccamo et al., 2020*; *Kühn et al., 2010*), both of which involve modification or synthesis of PG. But how a cytoplasmic bactofilin influences the PG cell wall in the periplasm is unclear. Bactofilins are identified by the presence of a bactofilin domain (DUF583), which forms a β-helix that is flanked by unstructured terminal regions. The function of the terminal regions and bactofilin domain of bactofilins are debated. It has been proposed that the terminal regions may be involved in modulating polymerization, binding to membranes and/or interacting with other proteins (*Deng et al., 2019*; *Kühn et al., 2010*; *Vasa et al., 2015*). However, experimental evidence to identify the function of the bactofilin domain and terminal regions of CcmA and other bactofilins is lacking.

Here, we performed structure–function analysis to determine how the bactofilin domain and the terminal regions of CcmA contribute to generating the helical shape of *H. pylori* cells by evaluating cell morphology, CcmA polymerization and filament bundling, and interaction with known Csd protein

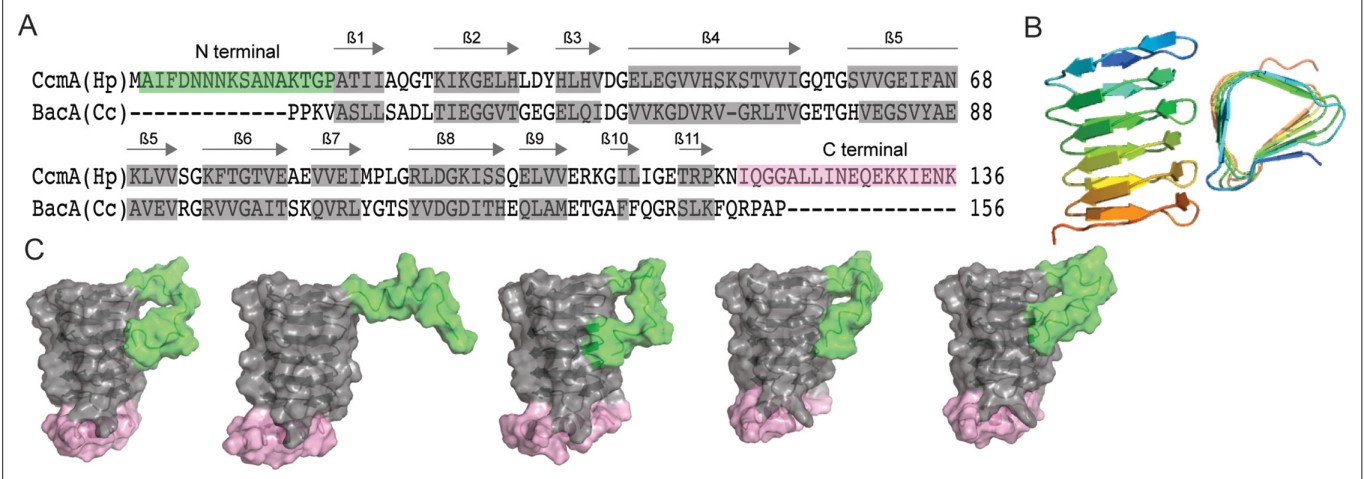

**Figure 1.** Predicted structure of CcmA. (**A**) Sequence alignment between BacA from *C. crescentus* and CcmA from *H. pylori* generated with SWISS-MODEL; gray boxes indicate β-strands formed in the bactofilin domain, the green box indicates the N-terminal region of CcmA, and the pink box indicates the C-terminal region of CcmA. (**B**) Model of the bactofilin domain of CcmA generated by RoseTTAFold, which forms a right-handed, three-sided, triangular β-helix. (**C**) Five models of CcmA generated in RoseTTAFold displaying the bactofilin domain (gray), N-terminal region (green), and C-terminal region (pink).

interaction partners. Additionally, we identified how protein binding partners impact CcmA localization. We found that interactions between CcmA and other Csd proteins are responsible for CcmA's ability to pattern the helical shape of *H. pylori* cells and identified that interactions between CcmA and its binding partners modulate CcmA localization and modulate protein stability of the PG-hydrolase Csd1. Together, these findings allow us to establish a new model for how *H. pylori*'s helical cell shape is generated. We clarify that CcmA participates in two separate helical-cell-shape-generating protein complexes. Additionally, these studies reveal that bactofilins can influence PG sacculus structure by modulating the protein stability of a PG hydrolase.

## Results

### CcmA is composed of a bactofilin domain and two short terminal regions

To facilitate structure–function analysis of CcmA's biochemical and cell biologic functions, we aligned the amino acid sequence of CcmA with BacA, a well-studied bactofilin from *Caulobacter crescentus* involved in stalk morphogenesis, using SWISS-MODEL (*Waterhouse et al., 2018*). This alignment revealed that CcmA is composed of a bactofilin domain (DUF 583) surrounded by two short terminal regions (*Figure 1A*). The N-terminal region is composed of amino acids 1–17, the bactofilin domain is composed of amino acids 18–118, and the C-terminal region is composed of amino acids 119–136. It has been proposed that bactofilins contain a short membrane binding motif within their N-terminal regions, which spans amino acids 3–12 in CcmA (*Deng et al., 2019*). However, whether the proposed membrane binding motif was functional in CcmA was unknown.

After identifying the boundaries between the bactofilin domain and the terminal regions of CcmA, we used RoseTTAFold (*Baek et al., 2021*), a protein structure prediction service that uses deep learning-based modeling methods, to predict the structure of CcmA. RoseTTAFold generated five models of CcmA. In all five, the bactofilin domain forms a three-sided, right-handed β-helix with a left-handed twist (*Figure 1B*), the N-terminal region is unstructured, and part of the C-terminal region forms a short α-helix and caps the end of the bactofilin domain (*Figure 1C*). In four of the five models, the first several amino acids of the N-terminal region interact with the bactofilin domain, suggesting that the N-terminal region could potentially block a surface of the bactofilin domain.

### The N-terminal region and bactofilin domain are together necessary and sufficient to pattern the helical shape of *H. pylori* cells

After identifying that CcmA was composed of an N-terminal region that may contain a membrane binding motif, a bactofilin domain, and a C-terminal region, we wanted to identify how each region of CcmA contributed to patterning the helical shape of *H. pylori* cells. The bactofilin literature suggested that the terminal regions on either end of the bactofilin domain may be involved in binding to membranes, modulating protein–protein interactions, or modulating polymerization (*Deng et al., 2019*; *Kühn et al., 2010*; *Vasa et al., 2015*), and we hypothesized deleting the terminal regions of *ccmA* would alter cell shape. Thus, we generated *H. pylori* strains that have truncated versions of *ccmA* at the native *ccmA* locus. We generated five strains, one strain missing the N-terminal region (*ΔNT*, missing amino acids 2–17), another strain missing the proposed membrane binding motif within the N-terminus (*ΔMM*, missing amino acids 3–12), a third strain lacking the five amino acids at the end of the N-terminus (*Δ13–17*, lacking amino acids 13–17), a fourth strain missing the C-terminal region (*ΔCT*, missing amino acids 119–136), and lastly, a strain expressing only the bactofilin domain that lacks both the N-terminal region and C-terminal region (BD, expressing amino acids 18–118) (*Figure 2A*). The open-reading frames (ORFs) of the end of *csd1* and the beginning of *ccmA* overlap by 82 nucleotides (*Sycuro et al., 2010*), thus to avoid impacting the function of Csd1 by removing several amino acids from the C-terminus when truncating the N-terminus of CcmA, we expressed a second copy of *csd1* at *rdx*A, a locus commonly used for complementation (*Smeets et al., 2000*). In previous work, we confirmed that both *ccmA* and *csd1* can be individually deleted and complemented back without impacting the expression of the other (*Sycuro et al., 2010*; *Yang et al., 2019*).

All strains expressed the truncated versions of CcmA measured by Western blotting with an α-CcmA polyclonal antibody; however, in strains lacking the C-terminus (*ΔCT* and BD), we saw much lower signal on the blot (*Figure 2B*). In follow-up experiments, we determined that the α-CcmA polyclonal

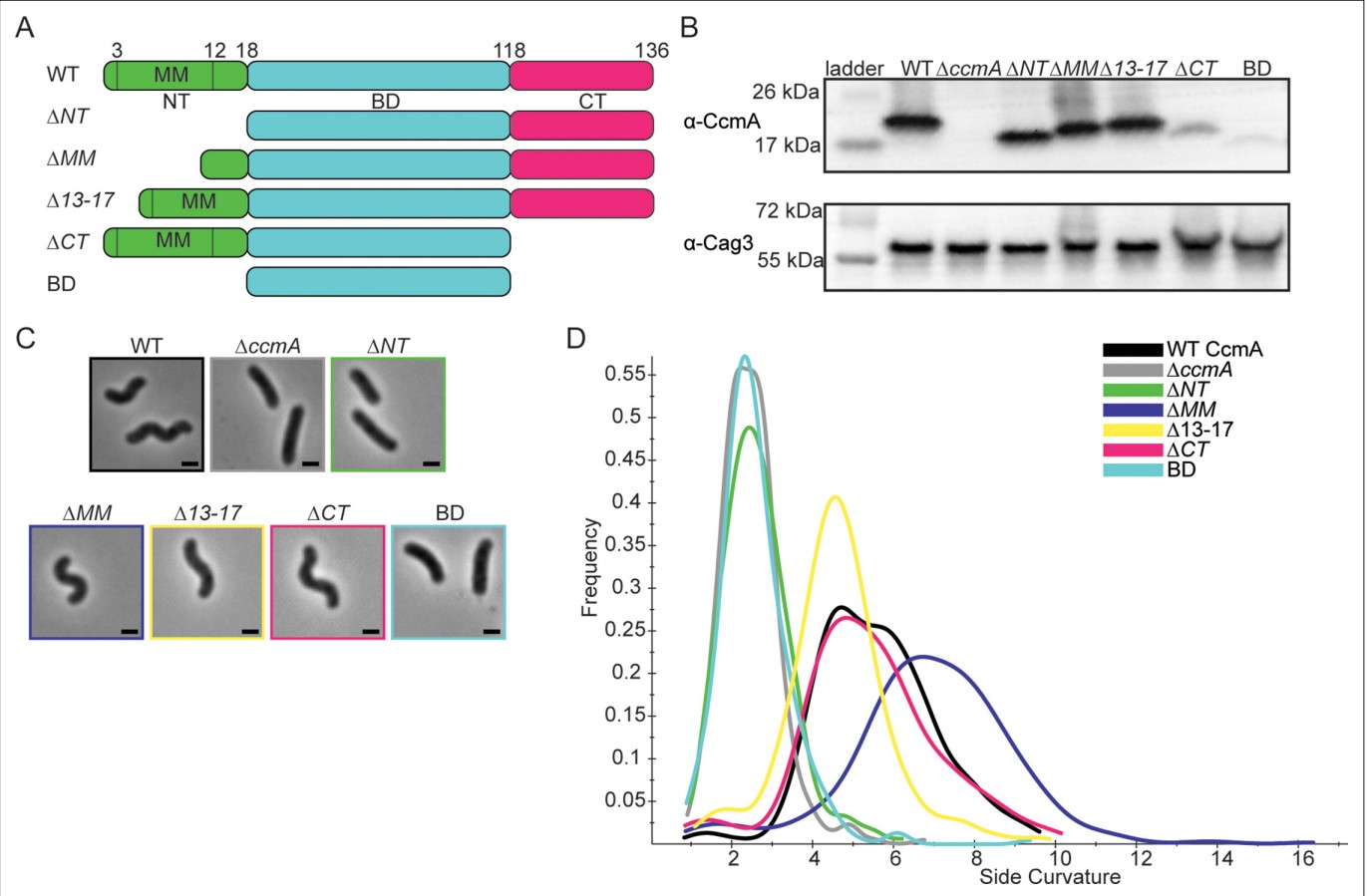

**Figure 2.** *H. pylori* cells expressing CcmA truncation mutants have different cell shapes. (**A**) Schematic of CcmA and truncation mutants. The N-terminal region (NT, green), bactofilin domain (BD, cyan), and C-terminal region (CT, pink) are indicated, as well as the putative membrane binding motif (MM). (**B**) Western blot to detect CcmA in whole-cell lysates of *H. pylori* strains expressing truncated versions of CcmA probed with α-CcmA polyclonal antibody, and α-Cag3 polyclonal antibody as a loading control. (**C**) Phase-contrast images of *H. pylori* strains expressing truncated versions of CcmA diagramed in panel (**A**), scale bars = 1 µm. (**D**) Histogram displaying the side curvature of *H. pylori* strains expressing truncated versions of CcmA (WT n = 476 cells, *ΔccmA* n = 397 cells, *ΔN* n = 275 cells, *ΔMM* n = 391 cells, Δ13–17 n = 361 cells, *ΔCT* n = 263 cells, BD n = 315 cells). Data are representative of two independent biological replicates. Strains used: SSH51A, SSH53A, SSH55A, SSH54A, SSH67A, SSH56B, and SSH57A.

The online version of this article includes the following source data and figure supplement(s) for figure 2:

**Source data 1.** Raw, unedited Western blots probed with α-Cag3 and α-CcmA antibodies in whole-cell lysates of *H. pylori* strains expressing truncated versions of CcmA.

**Figure supplement 1.** Robust CcmA detection with α-CcmA polyclonal antibody requires the C-terminus.

**Figure supplement 1—source data 1.** Raw, unedited Western blots probed with the α-CcmA antibody in purified CcmA.

**Figure supplement 2.** Purification of WT and mutant 6-his CcmA.

antibody fails to recognize versions of CcmA lacking the C-terminus (*Figure 2—figure supplement 1*). We ran a Western blot probed with our polyclonal α-CcmA antibody on purified 6-His WT CcmA and CcmA truncation mutants (*Figure 2—figure supplement 2*). Although equal concentrations of each purified protein were loaded, the polyclonal α-CcmA antibody poorly recognized versions of CcmA lacking the C-terminal region (*ΔCT* and BD), suggesting that the *ΔCT* and BD proteins are likely expressed in *H. pylori* cells at similar levels to other versions of CcmA but are not robustly recognized by the α-CcmA polyclonal antibody.

Next, after confirming that all truncated versions of CcmA were expressed, we assessed the shapes of the cells and found that strains lacking the N-terminal region (*ΔNT* and BD) were indistinguishable from *ΔccmA* cells (*Figure 2C and D*), while cells lacking the C-terminal region of *ccmA* resembled WT cells. Cells that lacked either the putative membrane binding motif (*ΔMM*) or the last five amino acids of the N-terminal region (*Δ13–17*) were helical but had slightly different side curvature values than the

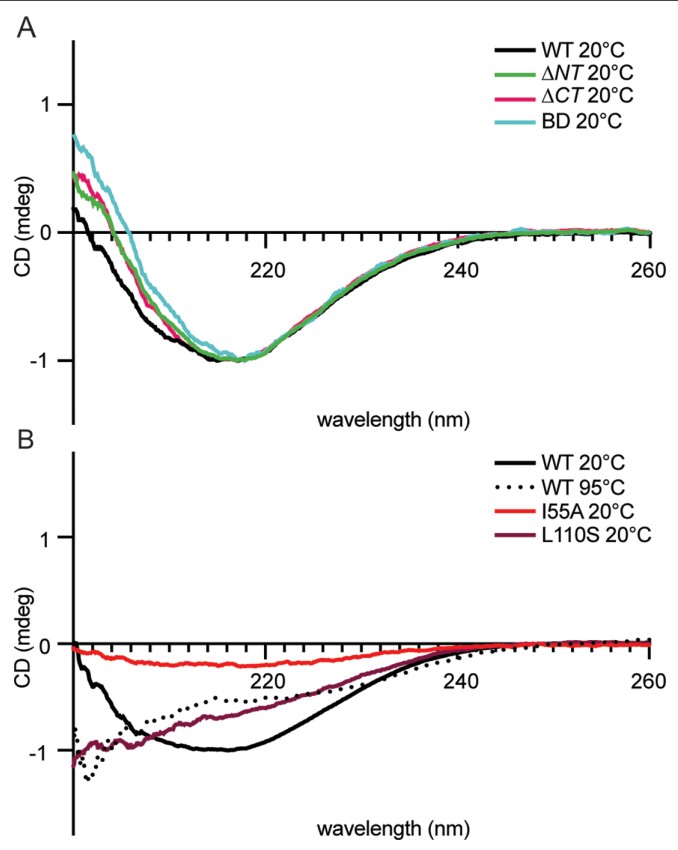

**Figure 3.** Circular dichroism spectra of purified 6-his CcmA suggests CcmA is composed of predominantly β-sheets. (**A**) Circular dichroism spectra of WT and truncated versions of CcmA at room temperature (20°C), WT (black), ΔNT (green), ΔCT (pink), and BD (cyan). (**B**) Circular dichroism spectra of WT CcmA at room temperature (20°C, black), melted (95°C, dotted line), and two point mutants (I55A, red, and L110S, burgundy).

WT population, suggesting that the N-terminal region of CcmA may have multiple functional regions within it.

## Circular dichroism spectroscopy reveals that CcmA truncation mutants maintain β-strand secondary structure

Considering the result that *H. pylori* strains lacking the N-terminal region of CcmA, cells resembled Δ*ccmA* mutants, we were curious whether loss of the terminal regions of CcmA might destabilize the protein and/or influence formation of the predicted β-strands (*Figure 1*). Circular dichroism (CD) spectroscopy can be used to identify the secondary structure and folding of proteins. Different structural elements have characteristic CD spectra, and proteins that consist of β-strands have negative peaks at 216 nm (*Greenfield, 2006*). We recombinantly expressed and purified 6-his CcmA (*Figure 2—figure supplement 2*) and collected CD spectra from 200 to 260 nm at fixed temperatures to identify the secondary structure and measure its thermal stability.

CD spectra of WT CcmA and the truncation mutants ΔNT, ΔCT, and BD at 20°C all look very similar; they have the characteristic negative peak at 216 nm, indicating β-strand secondary structure (*Figure 3A*). Together, these data, combined with the modeling by RoseTTAFold in *Figure 1C*, suggest that CcmA's bactofilin domain very likely forms a β-helix and confirms that neither terminal region is required for the bactofilin domain to fold properly. In contrast, we also ran CD scans on WT CcmA at 95°C and two CcmA point mutants we have previously shown were unable able to rescue the CcmA null cell shape defect, indicating that they were nonfunctional (*Taylor et al., 2020*) at 20°C. The CD spectra of both mutants (I55A and L110S) at 20°C resemble spectra of melted WT CcmA at 95°C (*Figure 3B*), suggesting that these proteins are not folded. Thus, while CcmA ΔNT, BD, I55A, and L110S all have the same cell shape phenotype when expressed in *H. pylori* – loss of helical cell shape

– the mechanism differs. In CcmA I55A and L110S, CcmA is not folded and nonfunctional, while in CcmA *ΔNT* and BD, the protein is folded, but loss of the N-terminus causes a loss of helical cell shape.

## The bactofilin domain alone is capable of polymerization: the terminal regions of CcmA modulate lateral interactions between CcmA filaments

Given that terminal region truncation mutant proteins appeared stable and displayed CD spectra indicative of a beta helical structure, we wondered whether their cell shape phenotypes correlated with altered polymerization properties. We assessed polymerization by negative stain and transmission electron microscopy (TEM) of purified truncated versions of CcmA and compared those to full-length protein. As previously reported (*Holtrup et al., 2019*; *Taylor et al., 2020*), WT CcmA polymerizes into small filaments that measure on average 3.04 nm (SD = 0.79, n = 18) in width. The filaments wrap together to form bundles of approximately 21.34 nm (SD = 4.82, n = 24) wide, and form lattice structures (*Figure 4A and G*). *ΔNT* formed filaments and bundles that were more homogenous, straighter, and thinner than WT and appear to only contain 2–4 filaments each. *ΔNT* failed to form lattice structures (*Figure 4B*). *ΔMM* resembled WT CcmA, forming filaments and bundles of approximately the same width as WT and formed lattices (*Figure 4C and G*). *Δ13–17* resembled *ΔNT* (*Figure 4D and G*), suggesting that amino acids 13–17 are responsible for filament bundling properties of *ΔNT*. *ΔCT* formed bundles that were wider than WT CcmA but failed to form lattices (*Figure 4E and G*). Lastly, BD formed straight bundles that were narrower than those found in WT samples and failed to form lattices (*Figure 4F and G*). These data suggest that the bactofilin domain is sufficient for polymerization of CcmA while the N- and C-terminal regions of CcmA regulate interpolymer interactions that impact the width of bundles by modulating the number of filaments that form a bundle, and the ability of CcmA to form lattices. Taken together with our experiments from *Figure 2*, where we expressed the truncated versions of CcmA in *H. pylori*, these data suggest that the ability of CcmA to polymerize is not sufficient to pattern the helical shape of *H. pylori* cells and in vitro lattice formation does not correlate with helical cell shape.

## CcmA's bactofilin domain mediates interactions with Csd5 and Csd7, and the N-terminal region of CcmA inhibits interactions with Csd7

Given that the truncated versions of CcmA were folded properly and were capable of polymerization in vitro, we hypothesized that the loss of helical cell shape phenotypes we observed when we deleted the N-terminal region of CcmA in *Figure 2* could be caused by disrupting interactions between CcmA and other cell-shape-determining proteins. CcmA was previously shown to interact with two Csd proteins, Csd5 and Csd7, by co-immunoprecipitation experiments. Csd5-CcmA interactions were readily detected by co-immunoprecipitation followed by mass spectrometry and Western blotting. However, Csd7-CcmA interactions could only be detected when a cross-linking reagent was included in co-immunoprecipitation experiments and mass spectrometry was used to detect CcmA, suggesting that Csd7-CcmA interactions are weak or potentially transient (*Blair et al., 2018*; *Yang et al., 2019*). Interactions between Csd5 and Csd7 were never detected by co-immunoprecipitation experiments followed by mass spectrometry in a WT or *ΔccmA* background (*Blair, 2018*; *Blair et al., 2018*; *Yang et al., 2019*). To probe how each region of CcmA contributes to interactions with Csd5 and Csd7, we performed co-immunoprecipitation experiments followed by Western blotting on strains expressing truncated versions of CcmA. Strains were generated that expressed truncated versions of *ccmA* at the native *ccmA* locus, *csd1* at the *rdxA* locus as described earlier, and either *csd5-2X-FLAG* or *csd7-3X-FLAG* at their native loci. We immunoprecipitated either Csd5-FLAG or Csd7-FLAG with α-FLAG agarose beads, then performed Western blotting and probed with the α-CcmA polyclonal antibody to identify whether the truncated versions of CcmA could still co-purify. When we immunoprecipitated Csd5-FLAG, all mutant versions of CcmA were detected by Western blotting of the IP fractions, except for the *ΔCT* mutant (*Figure 5A*), though it is likely that this mutant can still interact with Csd5-FLAG. Earlier we identified that the C-terminal region of CcmA is required for robust detection by our α-CcmA polyclonal antibody (*Figure 2—figure supplement 1*), and here, the BD mutant interacts with Csd5. Additionally, we tagged WT CcmA, *ΔCT,* and BD with a HaloTag in *H. pylori* at the native *ccmA* locus and performed a co-immunoprecipitation experiment to detect interactions with Csd5-FLAG (*Figure 5—figure supplement 1*). We found that both *ΔCT*-HaloTag and BD-HaloTag were expressed at comparable levels as CcmA-HaloTag and could be pulled down by Csd5-FLAG. These three pieces

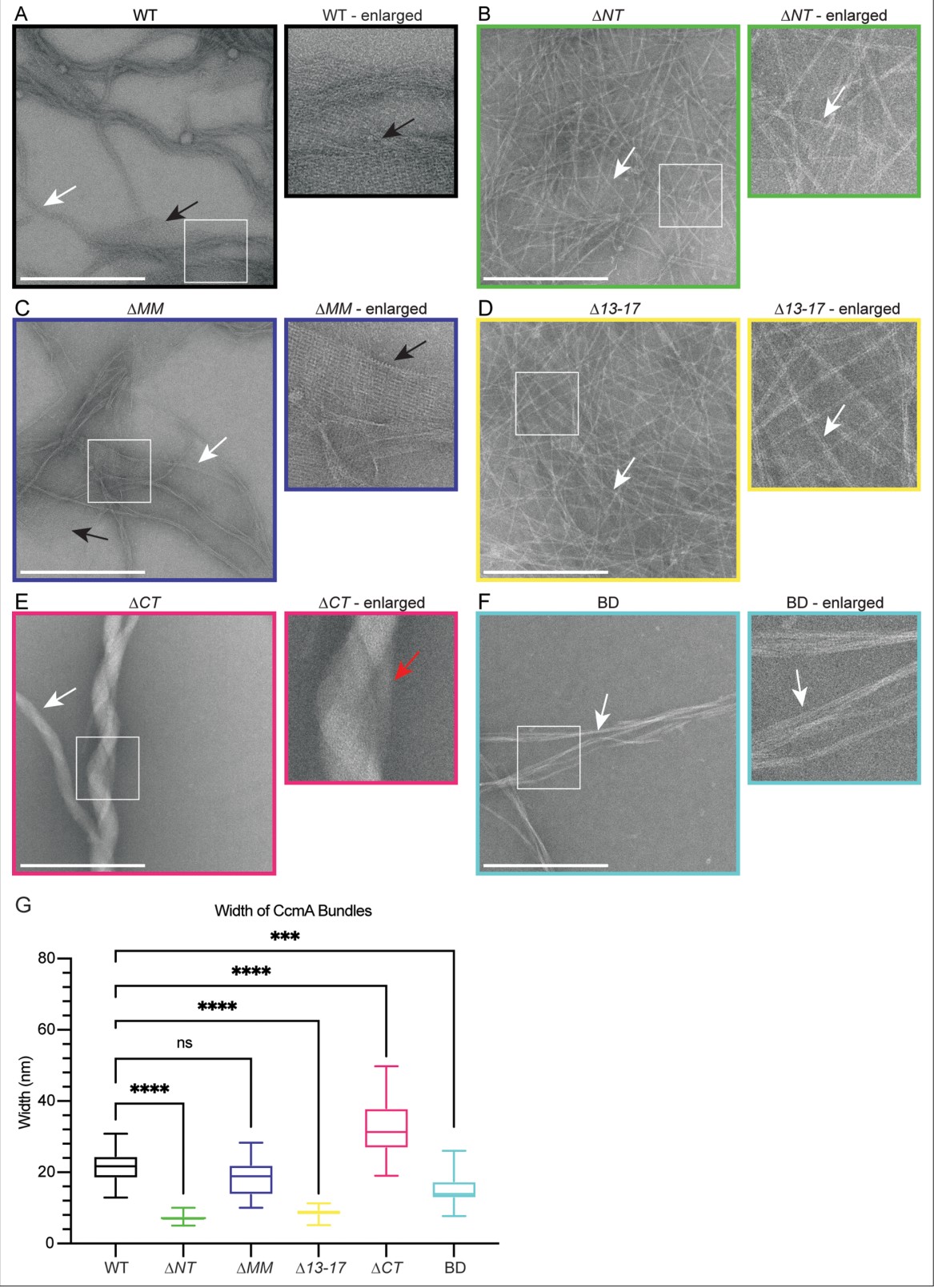

**Figure 4.** The bactofilin domain of CcmA is sufficient for in vitro polymerization while the terminal regions promote lateral polymer interactions. (**A–F**) Transmission electron microscopy (TEM) of negative stained, purified CcmA, scale bars = 500 nm. White boxes indicate region of micrograph that is enlarged in panel to the right of original. White arrows indicate bundles of CcmA, black arrows indicate lattice structures, and red arrow indicates two bundles wrapping around each other. Data are representative of two independent biological replicates. (**G**) Box and whiskers plot displaying width

*Figure 4 continued on next page*

*Figure 4 continued*

of purified CcmA bundles and filaments. Median, min, max, 25th percentile, and 75th percentile are displayed. One-way ANOVA, Dunnett's multiple-comparison test. ns p>0.5, ***p≤0.001, ****p≤0.0001. WT n = 24, *ΔNT* n = 29, *ΔMM* n = 25, *Δ13–17* n = 31, *ΔCT* n = 24, BD n = 32.

---

of evidence lead us to conclude that the *ΔCT* mutant can interact with Csd5, suggesting that only the bactofilin domain is required for interactions between CcmA and Csd5-FLAG.

When we immunoprecipitated Csd7-FLAG, we found that, as previously reported (*Yang et al., 2019*), WT CcmA was not able to be detected in the IP fractions by Western blotting above background levels. However, versions of CcmA lacking the N-terminal region were robustly co-purified by Csd7 (*Figure 5B*), suggesting that the N-terminal region inhibits WT CcmA from being co-immunoprecipitated by Csd7-FLAG.

To identify whether deletion of the N-terminus of CcmA caused CcmA, Csd7, and Csd5 to interact, we repeated Csd7-FLAG co-immunoprecipitation experiments in strains expressing WT CcmA, a strain lacking CcmA, and strains expressing CcmA *ΔNT* or CcmA BD, and probed for interactions with Csd5 by using a polyclonal antibody that recognizes the SH3 domain of Csd5 (*Blair et al., 2018*). We found that in all the conditions we tested Csd5 never co-purified with Csd7 (*Figure 5C*). Together, these data suggest that the bactofilin domain of CcmA mediates interaction with Csd5 and Csd7, the N-terminal region of CcmA inhibits interaction with Csd7, and CcmA forms separate complexes with Csd5 and Csd7. Our data also show that versions of CcmA lacking the C-terminal region are expressed at comparable levels to WT CcmA, although they are not recognized as robustly by the α-CcmA polyclonal antibody.

## Increased CcmA-Csd7 interactions preclude Csd1-Csd7 interactions, resulting in decreased Csd1 steady-state protein levels

After discovering that the N-terminal region of CcmA could inhibit interactions with Csd7 and that *ΔNT* CcmA was readily co-purified by Csd7, we reasoned that *ΔNT* CcmA could be binding to Csd7 and reducing its capacity to stabilize Csd1, causing decreased levels of Csd1 protein. This hypothesis was bolstered by the finding that *Δcsd7*, *Δcsd1*, and *ΔNT* cells all share the same curved rod shape phenotype (*Figure 4*; *Yang et al., 2019*). To test this hypothesis, we assayed Csd1 expression by Western blotting and probing with an α-Csd1 polyclonal antibody in whole-cell lysates of the *H. pylori* strain lacking the N-terminal region of *ccmA*. As discussed previously, this strain expresses two copies of *csd1*; there is an extra copy of *csd1* at *rdxA* that was added to account for potentially impacting Csd1's function when truncating *ccmA* (the ORFs of the end of *csd1* and the beginning of *ccmA* overlap by 82 nucleotides). As predicted, we found that in the strain lacking the N-terminal region (*ΔNT*) Csd1 levels are lower than in a strain expressing WT CcmA (*Figure 6A and B*).

After identifying that Csd1 levels were depleted when the N-terminal region of CcmA was missing, we sought to identify whether Csd7-Csd1 interactions were diminished when Csd7-CcmA interactions increased. We performed Western blotting and probed for Csd1 levels in the samples we collected during our Csd7-FLAG co-immunoprecipitation experiments (*Figure 5B*). We found that in *H. pylori* strains lacking the N-terminal region of CcmA, Csd7-Csd1 interactions are decreased in comparison to when WT CcmA is expressed (*Figure 6C*), suggesting that when the N-terminal region of CcmA is deleted CcmA-Csd7 interactions increase and inhibit Csd7 from stabilizing Csd1, causing decreased Csd1 levels. Our data suggests that Csd1 stability is indirectly regulated by CcmA via Csd7.

## CcmA requires binding partners for proper localization to the major helical axis

It has been hypothesized that bactofilins are able to interact with the cell envelope directly and sense specific curvature values (*Deng et al., 2019*; *Kühn et al., 2010*; *Vasa et al., 2015*). To determine whether binding partners are required to direct CcmA to the cell envelope or whether it can localize to the cell envelope on its own, we first created an *H. pylori* strain that we could use to visualize CcmA by 3D structured illumination microscopy (SIM). We engineered a strain with *ccmA-HaloTag* at the native *ccmA* locus and a second copy of WT *ccmA* at *rdxA*. We found that cells that expressed CcmA-HaloTag as the sole copy of CcmA were still helical but had slightly lower-side curvature than WT cells; adding a second copy of *ccmA* at *rdxA* generated cells with side curvature more similar to

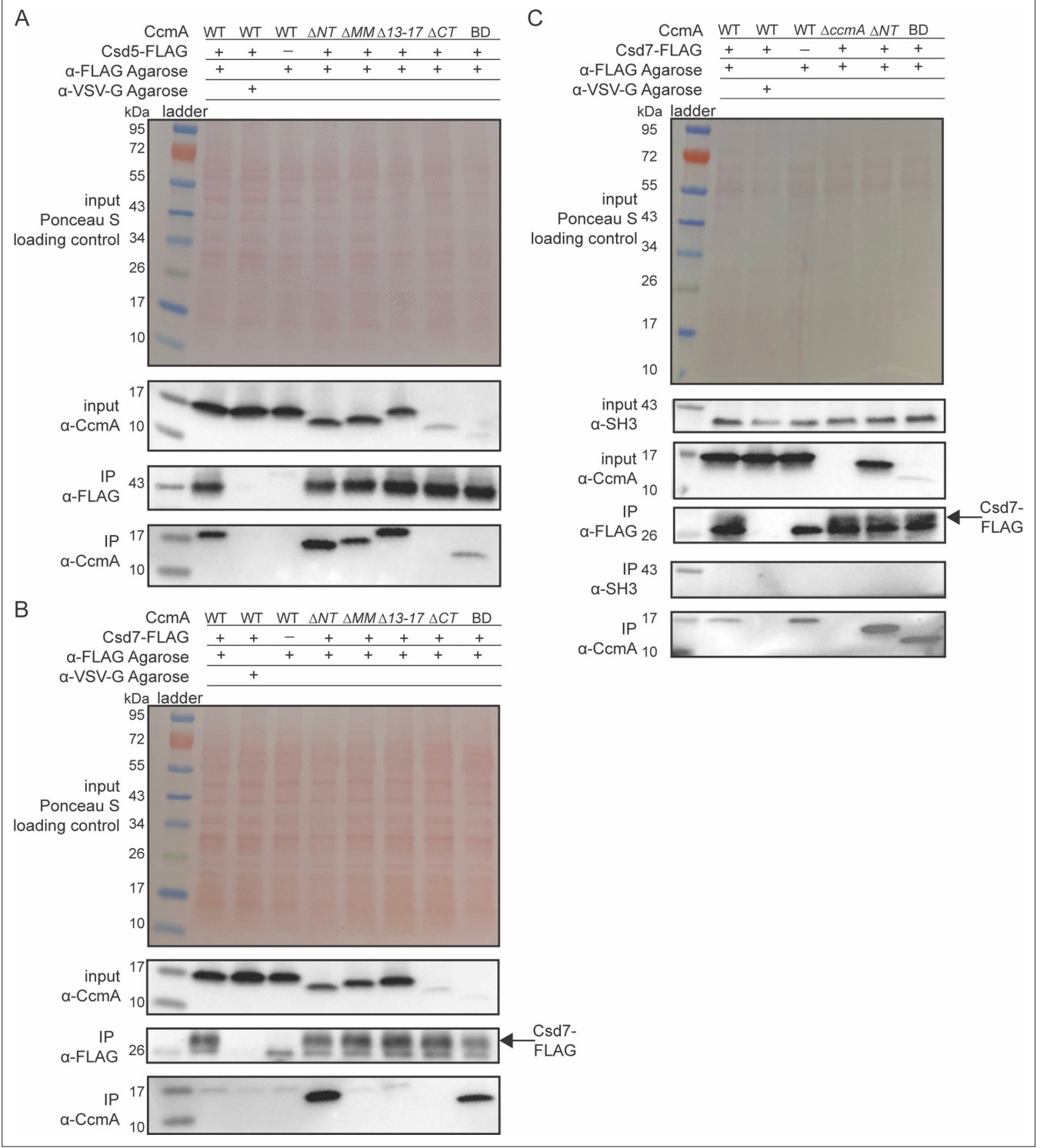

**Figure 5.** The bactofilin domain interacts with Csd5 and Csd7, and the N-terminal region inhibits Csd7 binding to CcmA. (**A, B**) Co-immunoprecipitation (co-IP) experiments to probe Csd5-2x-FLAG-CcmA interactions (**A**) and Csd7-3x-FLAG-CcmA interactions (**B**) in *H. pylori* cells are shown. Top row: Ponceau S staining of input fractions. Second row: Western blot probed with α-CcmA polyclonal antibody of input fractions. Third row: Western blot probed with α-FLAG monoclonal antibody of IP fractions. Bottom row: Western blot probed with α-CcmA polyclonal antibody of IP fractions. (**C**) Co-IP experiments to probe Csd7-3x-FLAG interactions with CcmA and Csd5 in *H. pylori* strains with and without CcmA. Top row: Ponceau S staining of input fractions. Second row: Western blot probed with α-SH3 polyclonal antibody of input fractions to detect the SH3 domain of Csd5. Third row: Western blot probed with α-CcmA polyclonal antibody of input fractions. Fourth row: Western blot probed with α-FLAG monoclonal antibody of IP fractions to detect Csd7-3x-FLAG. Fifth row: Western blot probed with α-SH3 polyclonal antibody to detect the SH3 domain of Csd5 in IP fractions. Bottom row: Western

*Figure 5 continued on next page*

*Figure 5 continued*

blot probed with α-CcmA polyclonal antibody of IP fractions. Data shown are representative of data from three independent biological replicates. Strains used: KHB157, LSH100, SSH59A, SSH58A, SSH68A, SSH60A, SSH65A, DCY71, DCY77, SSH79A, SSH78A, SSH82, SSH80A, and SSH81B.

The online version of this article includes the following source data and figure supplement(s) for figure 5:

**Source data 1.** Raw, unedited Western blots probed with α-CcmA and α-FLAG antibodies to detect CcmA and Csd5-FLAG or Csd7-FLAG in co-immunoprecipitation experiments.

**Figure supplement 1.** The C-terminus of CcmA-HaloTag is not required for interactions with Csd5.

**Figure supplement 1—source data 1.** Raw, unedited Western blots probed with α-HaloTag and α-FLAG antibodies to detect CcmA-HaloTag and Csd5-FLAG in co-immunoprecipitation experiments.

WT cells (*Figure 7—figure supplement 1A and B*). Next, to identify whether CcmA is capable of localizing to the cell envelope at its preferred curvature alone or whether localization is mediated by a binding partner, we deleted *csd5* and *csd7* individually and together in our *ccmA-HaloTag* strain. As an additional control, we deleted *csd6* in the *ccmA-HaloTag* strain. Csd6 is a PG carboxypeptidase that does not appear to interact with CcmA, Csd5, or Csd7 (*Blair et al., 2018*; *Sycuro et al., 2013*; *Yang et al., 2019*). Similar to Δ*csd5* mutants, Δ*csd6* mutants show a straight rod shape phenotype. We fixed and permeabilized cells, then labeled CcmA-HaloTag with a fluorescent ligand (JF-549) that binds to HaloTag and counterstained with fluorescent wheat germ agglutinin (WGA), which binds to PG, to label the cell envelope. We collected images of hundreds of cells from each population using 3D SIM and used software previously described to generate 3D reconstructions of individual cells from the WGA channel and calculate the Gaussian curvature at each point on the cell surface. Then we performed analysis to identify how much CcmA-HaloTag signal was at the cell envelope versus in the center of cells, and curvature enrichment analysis to investigate whether CcmA-HaloTag is enriched or depleted at specific Gaussian curvature values (*Bratton et al., 2019*; *Taylor et al., 2020*).

In WT cells, we found that CcmA-HaloTag signal is present as foci and small arcs that colocalize with the cell envelope (white signal) and there were some foci in the center of the cell (*Figure 7A*, top row). The CcmA-HaloTag signal closely resembled what we saw when we visualized CcmA-FLAG by immunofluorescence in previous work (*Taylor et al., 2020*). In cells lacking *csd6*, we found that CcmA-HaloTag signal is present at the cell envelope and there are some foci in the center of cells (*Figure 7A*, second row). In cells lacking *csd7*, cells are slightly curved and CcmA-HaloTag signal colocalizes with the cell envelope; however, there are also some foci in the center of the cell (*Figure 7A*, third row). In cells lacking *csd5*, there appears to be more CcmA-HaloTag signal in the center of cells than associated with the cell envelope (*Figure 7A*, fourth row). In cells lacking both *csd5* and *csd7*, we found that the cell shape resembles a Δ*csd7* mutant; however, there appears to be more CcmA-HaloTag signal in the center of the cells than in the Δ*csd7* mutant, similar to the Δ*csd5* mutant (*Figure 7A*, bottom row).

To quantify how much CcmA-HaloTag signal was associated with the cell envelope as compared to the amount of CcmA-HaloTag found in the center of the cells, we developed a method to measure the amount of CcmA-HaloTag signal that colocalized with the cell surface as we computationally generated sequential cell surface shells with smaller or larger diameters (*Figure 7B*, *Figure 7—figure supplement 1C and D*). We used the fluorescent WGA channel that labels the PG cell wall to generate the cell surface. As we computationally shrink the cell surface (offset from surface decreases), we observe the relative concentration of WGA and CcmA-HaloTag signal with respect to the position of the cell surface. In both a WT and Δ*csd5* cell, the WGA signal peaks at offset from surface = 0 nm where the PG cell wall is located within the periplasm. In WT cells, CcmA-HaloTag signal peaks near –50 nm from the PG cell wall and declines further to the inside of the cell. Based on cryo-electron tomography data, an approximate distance of 50 nm from the PG cell wall to the cytoplasm in *H. pylori* is expected (*Asmar et al., 2017*; *Chang et al., 2018*; *Kühn et al., 2010*; *Lu et al., 2022*; *Qin et al., 2017*). In contrast, in a Δ*csd5* cell, the CcmA-HaloTag signal peaks near –100 nm from the cell surface and continues to be high even as the surface of interest approaches center of the cell (approximately –300 nm), indicating that more CcmA-HaloTag signal is found near the center of the cell in Δ*csd5* cells than in WT cells. When we performed the same analysis on the entire populations of cells (292 WT and 326 Δ*csd5*), we found that while the overall trends were the same as when we analyzed individual cells, there was a lot of variability in the WT cell population. The variability of the WT population arises because many WT cells have both a population of CcmA-HaloTag at the cell

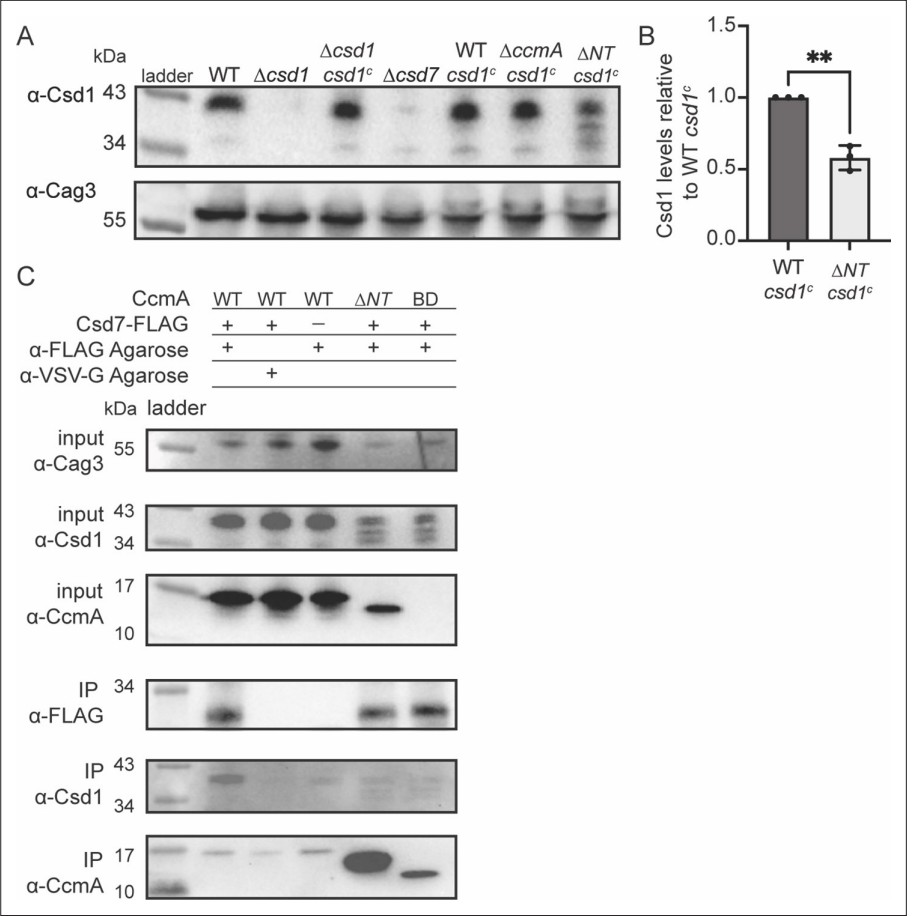

**Figure 6.** Csd1 expression is diminished when the N-terminal region of CcmA is deleted due to increased Csd7-CcmA interactions that preclude Csd7-Csd1 interactions. (**A**) Western blot of whole-cell extracts probed with α-Csd1 polyclonal antibody showing levels of Csd1; α-Cag3 polyclonal antibody was used as a loading control. Strains with 'csd1$^c$' indicate an extra copy of *csd1* was added at the *rdxA* locus. Data shown are representative of data from three independent biological replicates. (**B**) Quantification of Csd1 levels in the *ccmA ΔN csd1$^c$* strain relative to the WT *csd1$^c$*. Relative signal was calculated by dividing Csd1 signal by Cag3 signal for each replicate, then the data was normalized by dividing it by the value from the WT *csd1$^c$* strain from the same experiment. Data are pooled from three independent biological replicates. Unpaired *t*-test, **p≤0.001. (**C**) Co-immunoprecipitation (co-IP) experiments to probe Csd1 levels in relation to Csd7-3x-FLAG-CcmA interactions in *H. pylori* cells are shown. Top row: Western blot of input fractions probed with α-Cag3 polyclonal antibody as loading control. Second row: Western blot of input fractions probed with α-Csd1 polyclonal antibody. Third row: Western blot of input fractions probed with α-CcmA polyclonal antibody. Fourth row: Western blot probed with α-FLAG monoclonal antibody of IP fractions to detect Csd7-FLAG. Fifth row: Western blot probed with α-Csd1 polyclonal antibody of IP fractions. Bottom row: Western blot probed with α-CcmA polyclonal antibody of IP fractions. Data shown are representative of data from two independent biological replicates. Strains used: LSH100, LSH113, LSH121, DCY26, SSH51A, SSH53A, SSH55A, DCY71, SSH79A, and SSH81B.

The online version of this article includes the following source data for figure 6:

**Source data 1.** Raw, unedited Western blots probed with α-Cag3 and α-Csd1 antibodies in whole-cell lysates of *H. pylori* strains expressing truncated versions of CcmA.

envelope (peak at –50 nm) and in the center of the cell that causes a broader left tail in the population-level data (*Figure 7—figure supplement 1C and D*). In contrast, the *Δcsd5* population-level signal peak is broader and shifted towards the center of the cell (–80 nm). Thus, Csd5 is required for CcmA to localize to the cell envelope.

To identify where CcmA-HaloTag signal that is at the cell surface is in relation to cell curvature, we calculated the Gaussian curvature values of all cell surfaces represented in the populations. WT cells (which are helical) have a larger range of Gaussian curvature values on their cell surfaces (*Figure 7C*

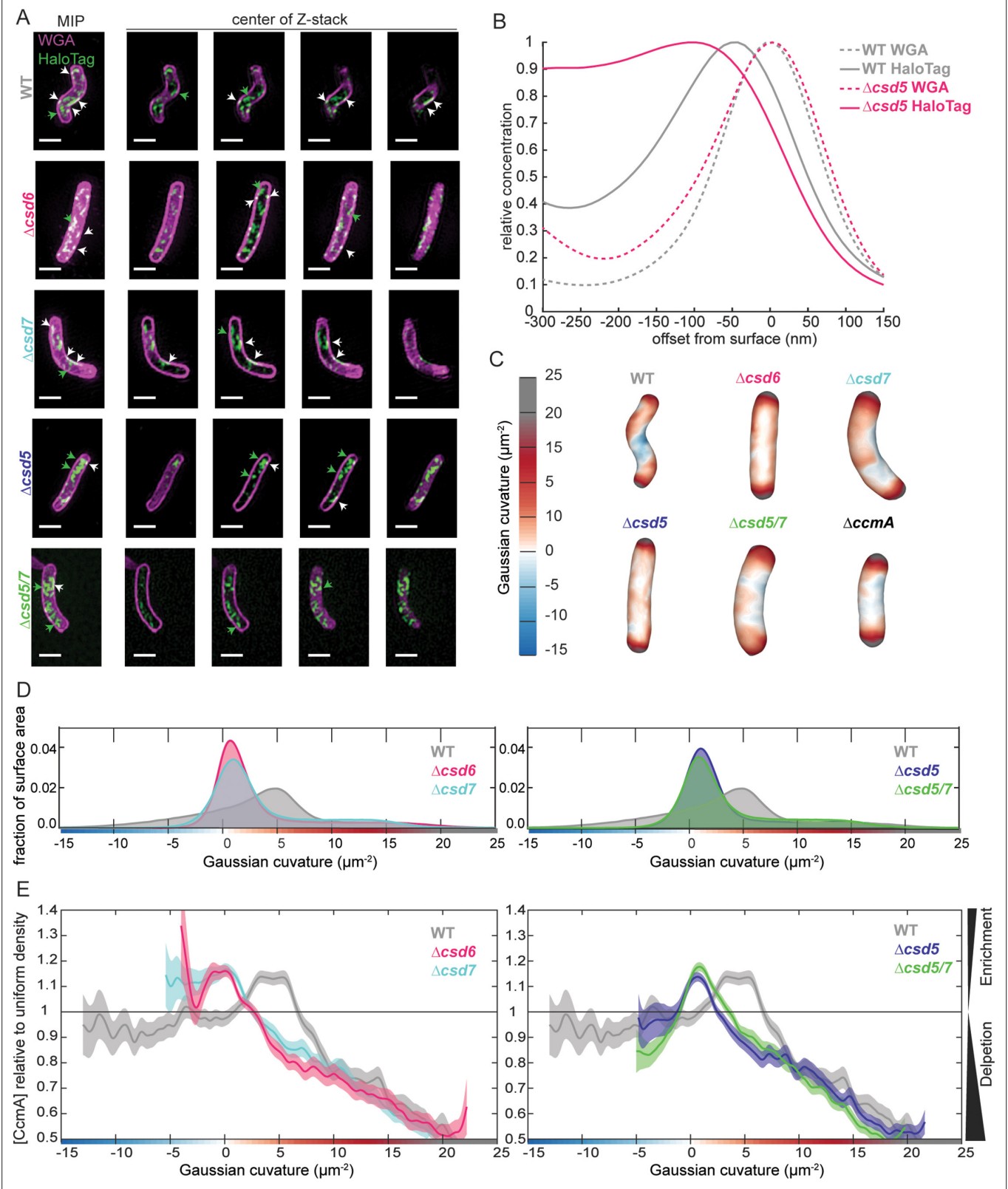

**Figure 7.** Csd5 is required for CcmA-HaloTag to localize to the cell envelope at the major helical axis. (**A**) Maximum intensity projections (MIPs) and frames from Z-stacks of *H. pylori* cells labeled with wheat germ agglutinin (WGA) to label the cell wall (magenta) and JF549 ligand to label CcmA-HaloTag (green), pixels are white where the two signals colocalize. Green arrows indicate cytoplasmic CcmA-HaloTag. White arrows indicate CcmA-HaloTag signal that colocalizes with the cell envelope. Scale bars = 1 µm. (**B**) Representative plots displaying the relative concentration of CcmA-

*Figure 7 continued*

HaloTag and WGA signal from one WT and one Δ*csd5* cell at the cell surface (offset from surface = 0) and at computationally generated cell surfaces that are inside (offset from surface < 0) and outside (offset from surface > 0) of the cell surface. (**C**) Representative *H. pylori* cells from each population with Gaussian curvature values mapped on the cell surface. (**D, E**) Top: histogram of Gaussian curvature of cell surfaces of each population. Bottom: surface Gaussian curvature enrichment of relative concentration of CcmA signal in each population where 1 = uniform concentration of CcmA, shaded regions indicate SEM. In panels (**C–F**), WT n = 292, Δ*csd6* n = 264 cells, Δ*csd7* n = 355, Δ*csd5* n = 326, Δ*csd5/7* n = 523. Strains used: SSH39B, SSH49A, SSH50A, JS09, SSH70A, and LSH117.

The online version of this article includes the following source data and figure supplement(s) for figure 7:

**Figure supplement 1.** Cell shape of CcmA-HaloTag strains and analysis of CcmA-HaloTag surface signal in populations.

**Figure supplement 2.** Cell shape and Csd1 levels of *ccmA ΔNT Δcsd5* mutants.

**Figure supplement 2—source data 1.** Raw, unedited Western blots probed with α-Cag3 and α-Csd1 antibodies in whole-cell lysates of *H. pylori* strains.

*and D*) compared to all other shape mutants. Interestingly, although all shape mutants are not helical, they do display different surface Gaussian curvature distributions and Δ*csd7* and Δ*csd5/7* populations look nearly identical, suggesting that Δ*csd5/7* double mutants resemble the shape of a Δ*csd7* mutant.

In WT cells, CcmA-HaloTag is enriched at the major helical axis (Gaussian curvature value of 5 ± 1 μm$^{-2}$; *Taylor et al., 2020*); enrichment peaks at +3.35 μm$^{-2}$ to +5.31 μm$^{-2}$ Gaussian curvature and is depleted at Gaussian curvature values lower than +0.97 μm$^{-2}$ and higher than +6.67 μm$^{-2}$. These data are consistent with previous experiments where CcmA-FLAG was detected by immunofluorescence (*Taylor et al., 2020*). In contrast, in all other strains, CcmA-HaloTag is depleted at regions of Gaussian curvature where CcmA-HaloTag is enriched in WT cells (+3.35 μm$^{-2}$ to +5.31 μm$^{-2}$), although those Gaussian curvature values are still present in all cell-shape-mutant populations (*Figure 7C D*). In Δ*csd6* and Δ*csd7* mutants, CcmA-HaloTag is enriched at negative Gaussian curvature values. In Δ*csd6* mutants, enrichment peaks at and continues to be enriched at all values lower than +0.11 μm$^{-2}$ Gaussian curvature (to –4 μm$^{-2}$ Gaussian curvature). In Δ*csd7* mutants, CcmA-HaloTag enrichment peaks at +0.15 μm$^{-2}$ and is enriched at all values below +0.15 μm$^{-2}$ (to –5.4 μm$^{-2}$ Gaussian curvature; *Figure 7E*). In contrast to Δ*csd6* and Δ*csd7* mutants, in Δ*csd5* mutants, CcmA-HaloTag is depleted at negative Gaussian curvature values (*Figure 7E*). Interestingly, in a Δ*csd5/7* double mutant, the CcmA-HaloTag enrichment pattern mirrors the Δ*csd5* mutant enrichment pattern. In Δ*csd5* mutants, CcmA-HaloTag is enriched at Gaussian curvature values from –1.18 μm$^{-2}$ to +2.5 μm$^{-2}$ and has a peak enrichment at +0.68 μm$^{-2}$, CcmA-HaloTag is depleted at all other values of Gaussian curvature. In a Δ*csd5/7* double mutant, CcmA-HaloTag is enriched at Gaussian curvatures of –0.99 μm$^{-2}$ to +3.46 μm$^{-2}$ and peaks at +0.85 μm$^{-2}$, CcmA-HaloTag is depleted at all other values of Gaussian curvature. Taken together, these data suggest that Csd5 directs CcmA to the cell envelope; CcmA is unable to localize to the cell envelope without Csd5. When helical cell shape is disrupted but Csd5 is present (in a Δ*csd6* or Δ*csd7* strain), CcmA localizes to regions of negative Gaussian curvature instead of the major helical axis. The finding that in a Δ*csd5*/Δ*csd7* double mutant the CcmA-HaloTag localization resembles a Δ*csd5* mutant indicates that Csd5, not Csd7, is the primary driver of CcmA localization in WT cells.

Given the requirement for Csd5 to localize CcmA to the cell envelope, we wondered what would happen in a Δ*csd5ccmAΔNT* strain. We could envisioned two possibilities. (1) Without Csd5, CcmA ΔNT would not be able to localize to the inner membrane and be close enough to interact with Csd7, thus CcmA ΔNT and Csd7 would not interact at high levels. In this case, Csd1 should be stable and the shape phenotype would resemble Δ*csd5* cells (straight rod). (2) Even though Csd5 is not present to direct CcmA ΔNT to the inner membrane, interactions between CcmA ΔNT and Csd7 are strong enough to direct CcmA to the inner membrane, thus CcmA ΔNT and Csd7 would still interact at high levels. In this case, we would expect diminished Csd1 protein levels and a cell shape phenotype similar to a Δ*csd5*Δ*csd1* (curved rod; *Sycuro et al., 2012*) and similar to Δ*csd5*Δ*csd7* shown in *Figure 7*. As shown in *Figure 7—figure supplement 2*, the Δ*csd5ccmAΔNT* strain has a curved rod phenotype and low levels of Csd1 supporting the second possibility, further reinforcing our hypothesis that Csd5 and Csd7 are part of separate protein complexes.

## Discussion

Multiple proteins have been identified that are required to generate the helical shape of *H. pylori* cells, several of which form protein complexes. However, the function of many of the individual proteins and complexes was unknown. Here, we sought to understand the function of the bactofilin CcmA in generating the helical shape of *H. pylori* cells. We identified the function of each of the regions/domains of CcmA in patterning the helical shape of *H. pylori* cells and clarified how the cell-shape-determining protein complexes in *H. pylori* function. This work provides mechanistic insight into how CcmA, and potentially other bactofilins, regulates cell shape and morphogenesis of bacteria.

In *H. pylori*, when the cytoplasmic bactofilin CcmA is lost, cells lose their helical shape and the PG muropeptide composition is altered (*Sycuro et al., 2010*). The PG sacculus is what ultimately controls the shape of bacterial cells, and it has been hypothesized that modification of PG cross-linking generates curvature and twist of the sacculus (*Huang et al., 2008*; *Sycuro et al., 2010*), but how loss of the cytoplasmic bactofilin CcmA could change the PG sacculus (in the periplasm) was unknown. Previously, we showed that CcmA and Csd7 (which is required for Csd1 protein stability) interact weakly and that Δ*ccmA*, Δ*csd1*, Δ*csd2*, and Δ*csd7* mutants all share the same curved rod cell shape phenotype and PG muropeptide composition (*Sycuro et al., 2010*; *Yang et al., 2019*). These observations led us to hypothesize that CcmA may indirectly regulate Csd1 activity and thus the PG cell wall, but a mechanism linking CcmA and Csd1 was unknown. In past work, we also identified that CcmA interacts with the transmembrane protein Csd5, which binds directly to PG, providing another potential link between CcmA and the PG sacculus. Additionally, we discovered that CcmA localizes to the major helical axis of *H. pylori* cells where rates of PG synthesis are also increased, suggesting that CcmA regulates PG synthesis in some fashion (*Blair et al., 2018*; *Taylor et al., 2020*; *Yang et al., 2019*). In this study, we built upon our previous body of work and identified several new mechanisms that clarify how CcmA regulates the helical cell shape, and thus the PG sacculus, of *H. pylori* by modulating interactions with Csd5 and Csd7.

We began this study by identifying how the bactofilin domain and short terminal regions of CcmA contributed to CcmA's ability to pattern helical cell shape in *H. pylori*. First, we identified that the N-terminal region of CcmA and the bactofilin domain are required for helical cell shape, but that the C-terminal region was not required. Next, in co-immunoprecipitation experiments, we found that the bactofilin domain of CcmA interacts with Csd5 and Csd7 and the N-terminal region inhibits the bactofilin domain from interacting with Csd7. When the N-terminal region of CcmA is removed, the bactofilin domain of CcmA binds to Csd7, which prevents Csd7 from stabilizing Csd1, causing Csd1 depletion. This provides the first potential mechanism by which CcmA regulates the PG hydrolase Csd1. Interestingly, when we modeled CcmA, in 4/5 models the N-terminal region of CcmA binds to the bactofilin domain of CcmA, potentially blocking one of the surfaces (*Figure 1C*). It is possible that by binding to the bactofilin domain the N-terminal region prevents interactions between the bactofilin domain of CcmA and Csd7. Future experiments will be needed to further clarify this mechanism.

We found that when other cell-shape-determining genes are deleted, CcmA can no longer localize to the major helical axis of the cell (Gaussian curvature value of $5 \pm 1$ $\mu m^{-2}$; *Taylor et al., 2020*), suggesting that CcmA is directed to its preferred location by a binding partner and is not capable of localizing to or sensing cell envelope curvature on its own. Loss of Csd5 causes substantially less CcmA to localize to the cell envelope, instead CcmA accumulates in the center of cells. The CcmA signal that does colocalize with the cell envelope in Δ*csd5* and Δ*csd5/7* cells is enriched near zero Gaussian curvature, which is the most common sidewall curvature value in these cells, and is depleted at all other Gaussian curvature values. When *csd6* or *csd7* are deleted, CcmA localizes to regions of negative Gaussian curvature. These data suggest that Csd5 recruits CcmA to the cell envelope. If helical cell shape is disrupted (due to loss of another cell shape protein) while Csd5 is still present, Csd5 directs CcmA to regions of negative Gaussian curvature. Additionally, the finding that CcmA localization phenotypes are the same in a Δ*csd5* mutant and a Δ*csd5/7* double mutant suggest that Csd5, not Csd7, controls CcmA localization. These results also support the hypothesis that there are two helical-cell-shape-generating protein complexes, one containing Csd5 and one containing Csd7.

Using what we have learned in this study, we can build a new model of the protein complexes that pattern the helical shape of *H. pylori* cells (*Figure 8*). In one protein complex, CcmA's bactofilin domain interacts with the cytoplasmic N-terminal region of the transmembrane protein Csd5 directing CcmA to the cell envelope. Csd5 brings MurF and PG precursors to the same region of the cell envelope

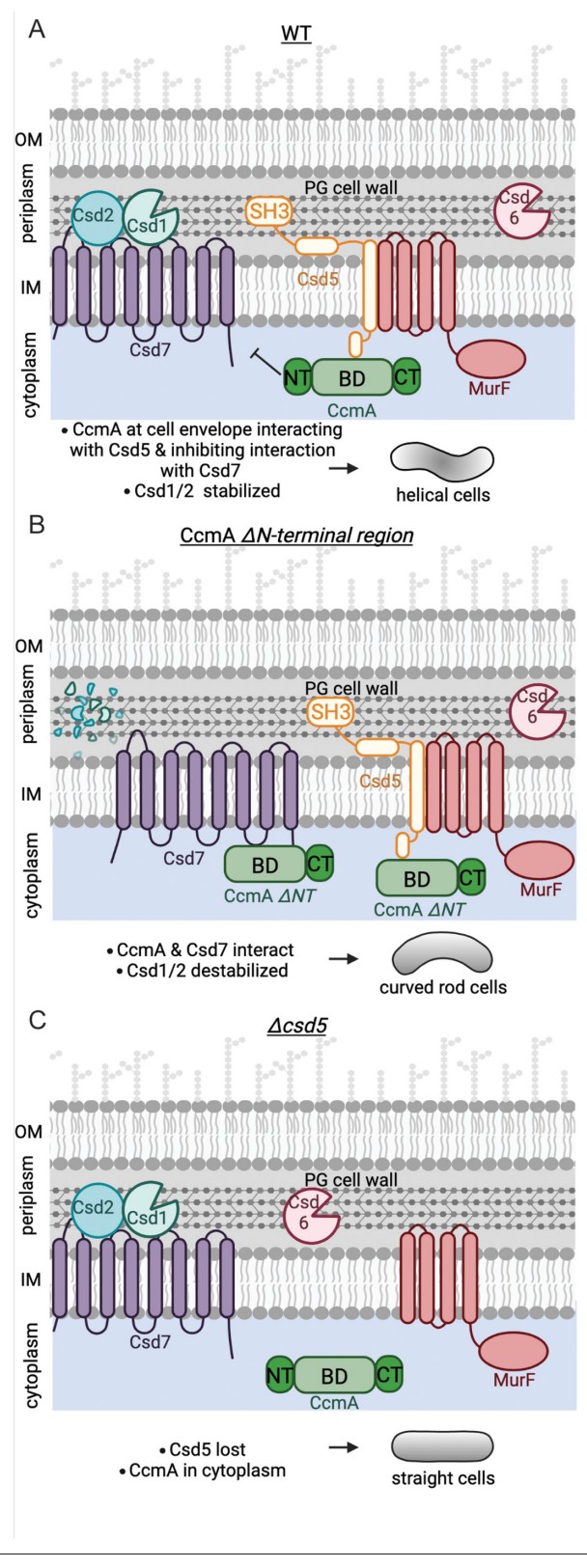

**Figure 8.** Schematic depicting CcmA's role in the helical cell shape complexes. There are two protein complexes, one containing Csd5, MurF, and CcmA, and another containing Csd7, Csd1, and Csd2. (**A**) In WT cells, Csd5 recruits WT CcmA to the cell envelope via CcmA's bactofilin domain. The N-terminal region of CcmA inhibits interaction with Csd7, allowing Csd1 to function and excluding Csd1 from the CcmA-Csd5-MurF complex. (**B**) In

*Figure 8 continued on next page*

*Figure 8 continued*

cells expressing CcmA *ΔNT*, Csd5 recruits *ΔNT* CcmA to the cell envelope via CcmA's bactofilin domain. The bactofilin domain of CcmA now also binds to Csd7 in a separate complex, inhibiting Csd7 from stabilizing Csd1. (**C**) When Csd5 is absent, CcmA cannot localize to the cell envelope, MurF is not directed to a particular location. OM, outer membrane; IM, inner membrane. This figure was created using Biorender.com.

for new PG to be inserted into the cell wall. CcmA's N-terminal region inhibits interaction of CcmA's bactofilin domain, which allows Csd7 to stabilize Csd1 (*Figure 8A*). When the N-terminus of CcmA is removed, interactions between Csd7 and CcmA are stabilized and Csd1 is no longer detected in the Csd7 complex (*Figure 8B*). Prior work had shown that Csd1 protein cannot be detected in a *Δcsd7* mutant, suggesting that Csd7 binding to Csd1 stabilizes this PG hydrolase. Steady-state levels of Csd1 are depleted in the *ccmA ΔNT* mutant, suggesting that Csd7 cannot simultaneously interact with Csd1 and CcmA. This finding explains why the *ccmA ΔNT* cells are curved rods, like a *Δcsd1* or *Δcsd7* cell; loss of the N-terminus of CcmA results in lower levels of Csd1, generating curved-rod-shaped cells. Thus, CcmA regulates Csd1 activity (cleaving tetra-pentapeptide crosslinks) spatially or temporally in relation to Csd5 and MurF to organize PG modification and insertion. In WT cells, spatial regulation of Csd1 protein levels by CcmA may act as a mechanism to prevent a futile cycle of PG insertion and hydrolysis and thus allow the increased synthesis at the major helical axis necessary to produce a helical cell. Our work raises the question of how Csd5 localizes to the major helical axis; Csd5 may sense curvature. Alternatively, the C-terminal SH3 domain of Csd5 can bind to PG directly (*Blair et al., 2018*) and may sense a particular curvature or structure of the cell wall.

In addition to understanding how the helical cell shape of *H. pylori* is controlled, this study provides information about how CcmA and potentially other bactofilins function. Modeling combined with CD data show that the bactofilin domain is a β-helix and the two short terminal regions are largely unstructured. Deletion of the terminal regions of CcmA does not impact the ability of the bactofilin domain to fold. Additionally, we identified that two previously studied CcmA point mutants that cause loss of function are unfolded (*Figure 3*). The I55 and L110 residues are homologous to residues mutated by Vasa and colleagues in BacA (V75 and F130 in BacA) and are predicted to reside in the interior of the hydrophobic core of the proteins and mutating these residues destabilizes the proteins likely through the hydrophobic effect (*Taylor et al., 2020*; *Vasa et al., 2015*). We also identified that the bactofilin domain is capable of polymerizing on its own, the N-terminal region promotes lateral interactions between filaments, and the N- and C-terminal regions are both necessary for lattice formation. These data correlate with what has been previously reported in the literature; in the bactofilin Ttbac from *Thermus thermophilus* loss of the N-terminal region (residues 1–10 in Ttbac) generated smaller, more ordered bundles of filaments (*Deng et al., 2019*), and it has been previously reported that purified *ΔCT* CcmA from *H. pylori* does not form lattice structures (*Holtrup et al., 2019*). Interestingly, in the case of CcmA *ΔNT* and BD, both purified proteins readily polymerize in vitro, yet are unable to pattern the helical cell shape of *H. pylori* when expressed in vivo, suggesting that CcmA has other functions besides being a purely structural element. For CcmA to function properly in the *H. pylori* cell-shape-determining protein complexes, it must polymerize, interact with Csd5, and inhibit interaction with Csd7. However, it is unclear what state CcmA is in when it interacts with Csd5 and inhibits interaction with Csd7; whether CcmA is polymerized or monomeric when it is in complex is unknown and will require further investigation .

In this study, we identified several putative functional motifs within CcmA. The N-terminal region (amino acids 1–17) is important for patterning helical cell shape and regulates interactions with Csd7, suggesting that the N-terminal region is a functional motif. Our data also suggest that amino acids 13–17 in the N-terminal region of CcmA promote lateral interactions between filaments in vitro and may be another functional motif. However, in CcmA, the putative membrane-binding motif (amino acids 3–12) suggested by Deng and colleagues (*Deng et al., 2019*) did not show a clear function in any of the experiments we conducted. In *H. pylori*, the putative membrane-binding motif of CcmA likely does not direct the protein to the inner membrane. Instead, interaction with Csd5 is required for CcmA to localize to the inner membrane. Interestingly, the C-terminal region of CcmA is not required for cell shape or interactions with Csd5 or Csd7 interactions, but it does have a role in regulating polymerization properties of purified CcmA.

Bactofilins are commonly found as part of complexes in other bacteria and help spatially organize cells to regulate where certain functions occur (*Sichel and Salama, 2020*; *Surovtsev and Jacobs-Wagner, 2018*). Our findings regarding how CcmA functions within a complex in *H. pylori* provide insight into how bactofilins operate within complexes in other bacterial species. Mechanisms similar to how CcmA functions in *H. pylori* have been reported in other systems; in *A. biprostecum,* a PG hydrolase, SpmX, directs the bactofilin BacA to the site of stalk synthesis, then BacA organizes the rest of the complex (*Caccamo et al., 2020*) to control stalk morphogenesis. Here, we identified that CcmA is not capable of localizing to the cell envelope without Csd5. While Csd5 is unique to the *Helicobacter* genus, transmembrane proteins in other species may also be essential for localization of bactofilins. Interestingly, the genomic arrangement of *csd1* (an M23 metallopeptidase) and *CcmA* overlapping is conserved in several other curved and helical species of Campylobacterota and Pseudomonadota (*Billini et al., 2019*; *Caccamo et al., 2020*; *Sycuro et al., 2010*), and has been reported in a spirochete (*Jackson et al., 2018*), suggesting that bactofilins may regulate M23 metallopeptidases to promote PG remodeling at specific regions of the cell envelope to control cell shape and morphogenesis of other morphological features. Our work is the first to suggest a mechanism by which a bactofilin can regulate a PG hydrolase indirectly through a transmembrane membrane protein and protein stability.

## Materials and methods

**Key resources table**

| Reagent type (species) or resource | Designation | Source or reference | Identifiers | Additional information |
|---|---|---|---|---|
| Antibody | Monoclonal α-FLAG M2 antibody produced in mouse | Sigma | Cat# F1804; RRID:AB_262044 | Used at 1:5000 for Western blot |
| Antibody | Polyclonal rabbit α-CcmA | *Blair et al., 2018* | | Used at 1:10,000 for Western blot |
| Antibody | Polyclonal rabbit α-SH3 | *Blair et al., 2018* | | Used at 1:5000 for Western blot |
| Antibody | Monoclonal mouse α-HaloTag | Promega | Cat# G9211; RRID:AB_2688011 | Used at 1:10,000 for Western blot |
| Antibody | Polyclonal rabbit α-Csd1 | *Yang et al., 2019* | | Used at 1:10,000 for Western blot |
| Antibody | Polyclonal rabbit α-Cag3 | *Pinto-Santini and Salama, 2009* | | Used at 1:20,000 for Western blot |
| Chemical compound, drug | α-FLAG M2 Affinity Gel | MilliporeSigma | Cat# A2220 | |
| Chemical compound, drug | Janelia Fluor HaloTag ligand 549 (JF-549) | Promega | Cat# GA1110 | |
| Chemical compound, drug | Wheat germ agglutinin, Alexa Fluor 488 Conjugate | Invitrogen | Cat# W11261 | |
| Other | ProLong Diamond Antifade Mountant | Invitrogen | Cat# P36961 | Mountant used for fluorescent microscopy |
| Commercial assay or kit | In-Fusion HD cloning kit | Takara | Cat# 638920 | |
| Commercial assay or kit | Zero Blunt TOPO PCR Cloning Kit, with One Shot Top10 Chemically Competent *E. coli* cells | Thermo Fisher Scientific | Cat# K280020 | |

### Bacterial strain construction and growth

Strains used in this work, as well as primers and plasmids used in strain construction, are available in *Tables 1–4*. *H. pylori* was grown in Brucella Broth (BD) supplemented with 10% heat-inactivated fetal bovine serum (Cytiva HyClone) without antimicrobials or on horse blood (HB) agar plates with micro-bials as described (*Sycuro et al., 2010*). For resistance marker selection, HB agar plates were supplemented with 15 µg/ml chloramphenicol, 25 µg/ml kanamycin, or 60 mg/ml sucrose. *H. pylori* strains were grown at 37°C under micro-aerobic conditions in a tri-gas incubator (10% $CO_2$, 10% $O_2$, and 80% $N_2$). For plasmid selection and maintenance in *Escherichia coli,* cultures were grown in Lysogeny broth (LB) or on LB-agar at 37°C supplemented with 100 µg/ml ampicillin or 25 µg/ml kanamycin.

**Table 1.** *H. pylori* strains used in this work.

| Strain | Relevant genotype or description | Reference or source |
|---|---|---|
| DCY26 | *csd7::catsacB* | *Yang et al., 2019* |
| DCY28 | *csd7::cat* | *Yang et al., 2019* |
| DCY71 | *csd7-3X-FLAG* | *Yang et al., 2019* |
| DCY77 | *csd7-3x-FLAG, ccmA::cat* | *Yang et al., 2019* |
| JS02 | *ccmA::ccmA-40 aa linker-HaloTag, rdxA::ccmA, csd5::cat2kan* | This work |
| JS09 | *ccmA::ccmA-40 aa linker-HaloTag, rdxA::ccmA, csd5::cat2kan, csd7::cat* | This work |
| JTH1 | *csd5-2X-FLAG cat* | *Blair et al., 2018* |
| JTH3 | *ccmA-2X-FLAG cat2kan* | *Blair et al., 2018* |
| KBH157 | *csd5-2X-FLAG, murF-3X-VSV-G::cat* | *Blair et al., 2018* |
| LSH100 | Wild-type *H. pylori*, NSH57 with *fliM* repaired | *Sycuro et al., 2010* |
| LSH108 | *rdxA::kansacB* | *Sycuro et al., 2010* |
| LSH113 | *csd1::cat* | *Sycuro et al., 2010* |
| LSH117 | *ccmA::catsacB* | *Sycuro et al., 2010* |
| LSH121 | *csd1::cat, rdxA::csd1* | *Sycuro et al., 2010* |
| LSH123 | *csd5::cat* | *Sycuro et al., 2012* |
| LSH148 | *ccmA::catsacB, rdxA::ccmA* | *Sycuro et al., 2010* |
| SSH31C | *ccmA::ccmA-12 aa linker-HaloTag* | This work |
| SSH33 | *ccmA::ccmA-40 aa linker-mNeonGreen* | This work |
| SSH38A | *ccmA::ccmA-40 aa linker-HaloTag* | This work |
| SSH39B | *ccmA::ccmA-40 aa linker-HaloTag, rdxA::ccmA* | This work |
| SSH41A | *ccmA::catsacB, csd7::csd7-3X-FLAG* | This work |
| SSH49A | *ccmA::ccmA-40 aa linker-HaloTag, rdxA::ccmA, csd5::cat* | This work |
| SSH50A | *ccmA::ccmA-40 aa linker-HaloTag, rdxA::ccmA, csd7::catsacB* | This work |
| SSH51A | *rdxA::csd1* | This work |
| SSH53A | *ccmA::catsacB, rdxA::csd1* | This work |
| SSH54A | *ccmA::ccmA ΔMM, rdxA::csd1* | This work |
| SSH55A | *ccmA::ccmA ΔNT, rdxA::csd1* | This work |
| SSH56B | *ccmA::ccmA ΔCT, rdxA::csd1* | This work |
| SSH57A | *ccmA::ccmA BD, rdxA::csd1* | This work |
| SSH58A | *ccmA::ccmA ΔMM, rdxA::csd1, csd5::csd5-2X-FLAG-cat* | This work |
| SSH59A | *ccmA::ccmA ΔNT, rdxA::csd1, csd5::csd5-2X-FLAG-cat* | This work |
| SSH60A | *ccmA::ccmA ΔCT, rdxA::csd1, csd5::csd5-2X-FLAG-cat* | This work |
| SSH65A | *ccmA::ccmA BD, rdxA::csd1, csd5::csd5-2X-FLAG-cat* | This work |
| SSH67A | *ccmA::ccmA Δ13–17, rdxA::csd1* | This work |
| SSH68A | *ccmA::ccmA Δ13–17, rdxA::csd1, csd5::csd5-2X-FLAG-cat* | This work |
| SSH70A | *ccmA::ccmA-40 aa linker-HaloTag, rdxA::ccmA, csd6::cat* | This work |
| SSH78B | *ccmA::ccmA ΔMM, rdxA::csd1, csd7::csd7-3X-FLAG* | This work |
| SSH79A | *ccmA::ccmA ΔNT, rdxA::csd1, csd7::csd7-3X-FLAG* | This work |
| SSH80A | *ccmA::ccmA ΔCT, rdxA::csd1, csd7::csd7-3X-FLAG* | This work |

*Table 1 continued*

| Strain | Relevant genotype or description | Reference or source |
| --- | --- | --- |
| SSH81B | *ccmA::ccmA BD, rdxA::csd1, csd7::csd7-3X-FLAG* | This work |
| SSH82 | *ccmA::ccmA Δ13–17, rdxA::csd1, csd7::csd7-3X-FLAG* | This work |
| SSH87A | *ccmA::ccmA-40 aa linker-HaloTag, csd5::csd5-2X-FLAG-cat* | This work |
| SSH89A | *ccmA::ccmA ΔCT-40 aa linker-HaloTag, rdxA::csd1,csd5::csd5-2X-FLAG-cat* | This work |
| SSH97A | *ccmA::ccmA BD-40 aa linker-HaloTag, rdxA::csd1, csd5::csd5-2X-FLAG-cat* | This work |
| SSH109A | *ccmA::ccmA ΔNT, rdxA::csd1, csd5::cat* | This work |
| TSH17 | *csd6::cat* | *Sycuro et al., 2013* |

All genetic manipulations of *H. pylori* were performed on the chromosome via natural transformation and allelic exchange by homologous recombination of either purified PCR products or plasmids. We used chloramphenicol (*cat*) and kanamycin (*aphA3*) resistance cassettes as markers in some strains (*Trieu-Cuot et al., 1985*; *Wang and Taylor, 1990*). In others, we used selectable and counter-selectable cassettes *catsacB* (chloramphenicol resistance and sucrose susceptibility) and *kansacB* (kanamycin resistance and sucrose susceptibility) for constructing marker-less strains (*Copass et al., 1997*).

All genomic DNA preparations from *H. pylori* were performed with the Wizard Genomic DNA Purification Kit (Promega). All plasmids were purified with the QIAprep Spin MiniPrep Kit (QIAGEN).

## Construction of *H. pylori* strains expressing truncated versions of *ccmA*

To construct strains expressing truncated versions of CcmA, we first generated a parent strain expressing an extra copy of *csd1* at *rdxA,* a locus commonly used for complementation (*Smeets et al., 2000*). First, we transformed LSH108 (*rxdA::kansacB*) with plasmid pLKS31 (*csd1* in pRdxA) to replace *kansacB* at *rdxA* with *csd1*, then selected for colonies that grew on HB agar plates (*Humbert and Salama, 2008*) containing sucrose but could not grow on HB plates containing kanamycin to generate strain SSH51A. Then, we replaced the *ccmA* gene with a *catsacB* cassette so that marker-less truncated versions of *ccmA* could be engineered by allelic exchange. To do this, we transformed strain SSH51A with genomic DNA from LSH117 (*ccmA::catsacB*) and selected for colonies that grew on HB plates containing chloramphenicol to generate strain SSH53A. We transformed strain SSH53A with purified PCR products generated by PCR SOEing (*Horton, 1995*) that had truncated versions of *ccmA* flanked by 800 nucleotides of homology upstream of the *ccmA* and 500 nucleotides of homology downstream of the *ccmA,* and selected for colonies that grew on HB plates containing sucrose, but not on HB plates containing chloramphenicol. After confirming that strains grew on selective plates, strains were confirmed by PCR and Sanger sequencing to sequence the *rdxA* and *ccmA* loci.

To generate the *ΔNT* strain (SSH55A, lacking amino acids 2–17), we used primers 'Csd1 F' and 'remove aa 2–17 R' to amplify the upstream PCR product and primers 'remove aa 2–17 F' and 'CcmA SDM dn R' to amplify the downstream PCR product from LSH100 (WT) genomic DNA. The resulting products were purified using the QIAquick PCR Purification Kit (QIAGEN). Then we stitched them together with PCR SOEing (*Horton, 1995*), gel extracted the product using the QIAquick Gel Extraction Kit (QIAGEN), and used the product directly for natural transformation of SSH53A. To generate the *ΔMM* strain (SSH54A, lacking amino acids 3–12), the same strategy was used, but we used primers 'remove aa 3–12 F' and 'remove aa 3–12 R' instead of 'remove aa 2–17 F' and 'remove aa 2–17 R.' To generate *Δ13–17* strain (SSH67A, lacking amino acids 13–17), the same strategy was

**Table 2.** *E. coli* strains used in this work.

| Strain | Relevant genotype or description | Reference or source |
| --- | --- | --- |
| BL21 (DE3) | Protein expression *E. coli* strain | New England Biolabs |
| Stellar | Cloning strain of *E. coli* | Takara |

**Table 3.** Plasmids used in this work.

| Plasmid | Genotype or description | Marker | Reference or source |
|---|---|---|---|
| pET15b | Modified pET15 vector (6-His expression vector) | Ampicillin | Barry Stoddard Lab, Fred Hutchinson Cancer Center |
| pKB62A | *6-His ccmA* in pET15b | Ampicillin | *Taylor et al., 2020* |
| pLC292 | pRdxA | Ampicillin | *Terry et al., 2005* |
| pLKS31 | *csd1* in pLC292 | Ampicillin | *Sycuro et al., 2010* |
| pSS15A | *6-His ccmA ΔMM* in pET15b | Ampicillin | This work |
| pSS16A | *6-His ccmA ΔCT* in pET15b | Ampicillin | This work |
| pSS17A | *6-His ccmA BD* in pET15b | Ampicillin | This work |
| pSS18B | *6-His ccmA Δ13–17* in pET15b | Ampicillin | This work |
| pSS19A | *6-His ccmA ΔNT* in pET15b | Ampicillin | This work |
| pCR Blunt II-TOPO vector | TOPO cloning vector | Kanamycin | Invitrogen |
| pKB69H | *6-His ccmA I55A* in pET15b | Ampicillin | *Taylor et al., 2020* |
| pKB72D | *6-His ccmA L110S* in pET15b | Ampicillin | *Taylor et al., 2020* |
| pFC30K | His6HaloTag T7 Flexi Vector | Kanamycin | Promega |
| pSS6-4 | *ccmA-12 aa linker-HaloTag* in pCR Blunt II-TOPO | Kanamycin | This work |
| pSS8E | *ccmA-40 aa linker-mNeonGreen* in pCR Blunt II-TOPO | Kanamycin | This work |
| pSS10-1 | *ccmA-40 aa linker-HaloTag* in pCR Blunt II-TOPO | Kanamycin | This work |

used, but with primers 'remove aa 13–17 2 F' and 'remove aa 13–17 2 R.' To generate the ΔCT strain (SSH56B, lacking amino acids 119–136), we followed the same strategy but used primers 'remove aa 119–136 F' and 'remove aa 119–136 R.'

To generate the BD strain (SSH57A, only amino acids 18–188), we amplified genomic DNA from strain SSH55A by PCR with primers 'Csd1 F' and 'remove aa 119–136 R' and genomic DNA from SSH56B with primers 'CcmA SDM dn R' and 'remove aa 119–136 F' and purified them using the QIAquick PCR Purification Kit (QIAGEN). Then we stitched them together with PCR SOEing (*Horton, 1995*), gel extracted the product using the QIAquick Gel Extraction Kit (QIAGEN), and used the product directly for natural transformation of SSH53A.

To generate strain SSH109A, a strain lacking *csd5* and the N-terminus of *ccmA*, strain SSH55A (*ccmA::ccmA ΔNT, rdxA::csd1*) was transformed with genomic DNA from LSH123 (*csd5::cat*). Colonies that grew on HB plates containing chloramphenicol were selected.

## Construction of *H. pylori* strains expressing FLAG-tagged versions of Csd5 or Csd7

To construct strains expressing Csd5-2X FLAG, strains with truncated versions of *ccmA* (SSH54A, SSH55A, SSH56B, SSH57A, SSH67A) were transformed with genomic DNA from JTH1 (contains a cat cassette downstream of *csd5-2X FLAG* at the native *csd5* locus). Colonies that grew on HB plates containing chloramphenicol were selected and confirmed by PCR. These are strains SSH58A, SSH59A, SSH60A, SSH65A, and SSH68A.

To construct strains expressing Csd7-3X FLAG, DCY71 (Csd7-3X-FLAG) was transformed with genomic DNA from LSH117 to replace *ccmA* with *catsacB* and clones that grew on HB plates supplemented with chloramphenicol were selected to generate strain SSH41A. Then, SSH41A was transformed with genomic DNA from strains SSH54A, SSH55A, SSH56B, SSH57A, or SSH67A to replace *catsacB* with truncated versions of *ccmA*, clones that grew on HB plates supplemented with sucrose but not chloramphenicol were selected. Next, strains were transformed with genomic DNA from LSH108 to place *kansacB* at the *rdxA* locus, clones that grew on HB plates supplemented with kanamycin were selected. Finally, clones were transformed with genomic DNA from SSH51A to replace *kansacB* with *csd1* at *rdxA*, clones that grew on HB plates supplemented with sucrose but not on HB

**Table 4.** Primers used in this work.

| Name | Sequence 5′ to 3′ |
| --- | --- |
| **Construction of *H. pylori* strains** | |
| Csd1 F | GAGTCGTTACATTAATGTGCATATCT |
| CcmA SDM dn R | GCTCATTTGAGTGGTGGGAT |
| Remove aa 2–17 F | CAATAAAGAAAGGAGCATCAGATGGCGACTATCATCGCTC |
| Remove aa 2–17 R | GAGCGATGATAGTCGCCATCTGATGCTCCTTTCTTTATTG |
| Remove aa 3–12 F | AATAAAGAAAGGAGCATCAGATGGCAGCAAAAACAGG |
| Remove aa 3–12 R | CCTGTTTTTGCTGCCATCTGATGCTCCTTTCTTTATT |
| Remove aa 119–136 F | GGGAAACTCGCCCTAAGAATTAGGGAATGATCCAATCTAG |
| Remove aa 119–136 R | CTAGATTGGATCATTCCCTAATTCTTAGGGCGAGTTTCCC |
| Remove aa 13–17 2 F | ATAACAATAATAAATCGGCTAATGCGACTATCATCGCTCA |
| Remove aa 13–17 2 R | TGAGCGATGATAGTCGCATTAGCCGATTTATTATTGTTAT |
| CcmA_40aa_linker_upstream_R | ACCTTGTCCGCTACCCTCAAGTTTATTTTCAATTTTCTTTTC |
| CcmA_mNeonGreen_dnstrm_F | GATGGATGAATTATATAAATAATAGGGAATGATCCAATCTAGTCT |
| CcmA_12_aa_link_Halo_R | AGTACCGATTTCACTACCACCACCACTACCACCACCACTACCACCACCTTTATTTTCAAT |
| CcmA_Halo_dnstrm_R | AGACTAGATTGGATCATTCCCTAACCGGAAATCTCCAGA |
| CcmA_12_aa_link_Halo_F | ATTGAAAATAAAGGTGGTGGTAGTGGTGGTGGTAGTGGTGGTGGTAGTGAAATCGGTACT |
| 40 aa linker halotag R | TGGAAAGCCAGTACCGATTTCGCCTTGACCTGGGCCAGATCC |
| 40 aa linker halotag F | GGATCTGGCCCAGGTCAAGGCGAAATCGGTACTGGCTTTCCA |
| C1_828 | GATATAGATTGAAAAGTGGAT |
| C1_829 | TTATCAGTGCGACAAACTGGG |
| **Cloning for protein purification** | |
| CcmA_pET15b_BamHI | GTTAGCAGCCGGATCCCTATTTATTTTCAATTTTCTTTTCTTGCTCATTGA |
| pET15b_ccmA_delta_aa2to17_NdeI | CGCGCGGCAGCCATATGATGGCGACTATCATCGCTCAAG |
| CcmA_pET15b_BamHI | GTTAGCAGCCGGATCCCTATTTATTTTCAATTTTCTTTTCTTGCTCATTGA |
| pET15b_ccmA_delta_aa3to12_NdeI | CGCGCGGCAGCCATATGATGGCAGCAAAAACAGGACCAG |
| CcmA_pET15b_NdeI | CGCGCGGCAGCCATATGATGGCAATCTTTGATAACAATAATAAATCGGCT |
| pET15b_ccmA_delta_aa119to136_BamHI | GTTAGCAGCCGGATCCCTAATTCTTAGGGCGAGTTTCC |
| pET15b_ccmA_delta_aa13to17_NdeI | CGCGCGGCAGCCATATGATGGCAATCTTTGATAACAATAATAAATCGGCT |

*Table 4 continued on next page*

*Table 4 continued*

| Name | | Sequence 5' to 3' |
|------|--|-------------------|

**Sequencing primers**

| Target | Primer name | |
|--------|-------------|--|
| *rdxA* | 1318 | GAAGCGGTTACAATCATCACGCCC |
| *rdxA* | 1319 | GCTTGAAAACACCCCTAAAAGAGCG |
| *ccmA* | 1432 | GATTACCATTTGCATGTAGATGGCG |
| *ccmA* | 1433 | CTAGAGATCTTACCATCAAGGCGC |
| *csd5* | Csd5-c-term-seq | AAGCGTGAAGGTTTTAGAAATCCA |
| *csd5* | 1194_R | CCACAAGCTCATCATCTTCCAAAA |
| pET15b | T7 F | CGAAATTAATACGACTCACTATAGG |
| pET15b | T7 R | CCTCAAGACCCGTTTAGAGGCC |

plates supplemented with kanamycin were selected and confirmed by PCR and Sanger sequencing of the *rdxA* and *ccmA* loci. Confirmed strains are SSH78B, SSH79A, SSH80A, SSH81B, and SSH82.

## Construction of *H. pylori* strains expressing CcmA-HaloTag

We used a flexible 40 amino-acid-long linker to link a HaloTag to the C-terminus of CcmA. To do this, we first constructed a strain where the C-terminus of *ccmA* was fused to *mNeonGreen* with the 40 amino-acid-long linker between the two, then modified it to replace *mNeonGreen* with *HaloTag*. We used a gBlock (IDT) that contained the linker followed by a codon-optimized version of *mNeonGreen* and used PCR SOEing (*Horton, 1995*) to generate a construct that had *ccmA-linker-mNeonGreen* flanked with 800 nucleotides of homology upstream of the *ccmA* and 500 nucleotides of homology downstream of the *ccmA*. We amplified the upstream portion of the construct from LSH100 (WT) genomic DNA with primers 'Csd1 F' and 'CcmA-40aa_linker_upstream_R' and the downstream portion with primers 'CcmA-mNeonGreen_dnstrm_R' and 'CcmA SDM dn R.' After cleaning up the PCR products with QIAquick PCR Purification Kit (QIAGEN), we performed PCR SOEing (*Horton, 1995*) to stitch together the two PCR products and the gBlock, the product was gel extracted using QIAquick Gel Extraction Kit (QIAGEN), then TOPO cloned using the Zero Blunt TOPO PCR Cloning Kit (Invitrogen). The plasmid was then used as a template to transform LSH117 for allelic exchange to replace the *catsacB* cassette at the *ccmA* locus with the construct. Clones that grew on HB plates containing sucrose but not chloramphenicol were selected and confirmed by PCR and Sanger sequencing to generate strain SSH33.

Next, we constructed a strain where HaloTag was fused to the C-terminus of *ccmA* with a 12-amino-acid-long linker. To do this, we generated a construct containing *ccmA-12aa-linker-HaloTag* flanked by 800 nucleotides of homology upstream of the *ccmA* and 500 nucleotides of homology downstream of the *ccmA*. We amplified PCR products from genomic DNA from LSH100 with primers 'Csd1 F' and 'CcmA_12_aa_link_HaloR' and primers 'CcmA_Halo_dnstrm_R' and 'CcmA_SDM_dn_R.' Next, we amplified a PCR product containing HaloTag from plasmid pFC30K (Promega) with primers 'CcmA_12_aa_link_Halo_F' and 'CcmA_Halo_dnstrm_R.' After cleaning up the PCR products with QIAquick PCR Purification Kit (QIAGEN), we performed PCR SOEing (*Horton, 1995*) to stitch the three PCR products together and TOPO cloned the purified PCR product using the Zero Blunt TOPO PCR Cloning Kit (Invitrogen). The plasmid was then used as a template to transform LSH117 and replace the *catsacB* cassette at the *ccmA* locus with the construct to generate strain SSH31C. Clones that grew on HB plates containing sucrose but not chloramphenicol were selected and confirmed by PCR and Sanger sequencing.

Finally, to construct a strain expressing *ccmA-40 aa-linker-HaloTag* at the native *ccmA* locus, we amplified a PCR product with primers 'Csd1 F' and '40 aa linker halotag R' from SSH33 genomic DNA and another PCR product with primers '40 aa linker halotag F' and 'CcmA SDM dn R' from SSH31C genomic DNA. After cleaning up the PCR products with QIAquick PCR Purification Kit (QIAGEN), we

performed PCR SOEing (*Horton, 1995*) to stitch the two products together, then gel extracted using QIAquick Gel Extraction Kit (QIAGEN), and TOPO cloned using the Zero Blunt TOPO PCR Cloning Kit (Invitrogen). The plasmid was then used to transform LSH117 and replace the *catsacB* cassette at the *ccmA* locus with the construct. Clones that grew on HB plates containing sucrose but not chloramphenicol were selected and confirmed by PCR and Sanger sequencing (SSH38A). We noticed that these strains had altered morphology (expanded view *Figure 4*), so to adjust the cell shape to be more like WT, we added a second copy of WT *ccmA* at *rdxA* by transforming LSH148 (*ccmA::-catsacB, rdxA::ccmA*) with genomic DNA from SSH38A to replace *catsacB* cassette with *ccmA-40 aa linker- HaloTag* to generate strain SSH39B. Clones that grew on HB plates containing sucrose but not chloramphenicol were selected and confirmed by PCR.

To generate strains lacking *csd5*, *csd7*, and *csd6*, SSH39B was transformed with genomic DNA from either LSH123 (*csd5::cat*), DCY26 (*csd7::cat*), or TSH17 (*csd6::cat*). Clones that grew on HB plates supplemented with chloramphenicol were selected to generate strains SSH49A, SSH50A, and SSH70A.

To generate JS09, a strain lacking *csd5* and *csd7*, we converted the *cat* cassette to a kanamycin resistance cassette in SSH49A (*ccmA-40 aa linker-HaloTag, csd5::cat, rdxA::ccmA*) by transforming it with a PCR product containing a kanamycin resistance cassette flanked by homology to a *cat* cassette. The cassette was amplified by PCR using primers 'C1_828' and 'C2_829' from JTH3, then the product was purified with the QIAquick PCR Purification Kit. After transforming SSH49A with the purified PCR product, we selected colonies that grew on HB plates supplemented with kanamycin but not chloramphenicol to generate strain JS02. Next, JS02 was transformed with genomic DNA from DCY28 (*csd7::cat*) and colonies that grew on HB plates supplemented with chloramphenicol and kanamycin were selected.

## Construction of vectors for recombinant expression of 6-his CcmA

Truncated versions of CcmA were cloned into pET15b, a cloning vector used for expressing N-terminal 6-his tagged proteins, with the In-Fusion cloning kit (Takara). First, pET15b was linearized with restriction enzymes BamHI-HF (NEB) and NdeI (NEB), then gel extracted and purified using QIAquick Gel Extraction Kit (QIAGEN). Next, PCR products containing truncated versions of CcmA flanked by 15 nucleotides of homology to the plasmid backbone were amplified and the products were purified with the QIAquick PCR Purification Kit (QIAGEN).

The *ΔNT* product was amplified from SSH55A genomic DNA with primers 'CcmA-pET15b_BamH1' and 'pET15b_ccmA_delta_aa2to17_NdeI.' The *ΔMM* product was amplified from SSH54A genomic DNA with primers 'CcmA-pET15b_BamH1' and 'pET15b_ccmA_delta_aa3to12_NdeI.' The *Δ13–17* product was amplified from SSH67A genomic DNA with primers 'CcmA-pET15b_BamH1' and 'pET15b_ccmA_delta_aa13to17_NdeI.' The *ΔCT* product was amplified from SSH56B genomic DNA with primers 'CcmA-pET15b_NdeI' and 'pET15b_ccmA_delta_aa119to136_BamHI.' The *BD* product was amplified from SSH57A genomic DNA with primers 'pET15b_ccmA_delta_aa2to17_NdeI' and 'pET15b_ccmA_delta_aa119to136_BamHI.'

After cloning following the manufacturer's instructions, colonies were verified by restriction digest and Sanger sequencing. Confirmed plasmids were transformed into BL21 DE3 (NEB) for recombinant expression of 6-his-CcmA.

## Protein structure prediction

SWISS-MODEL was used to identify the boundaries between the bactofilin domain and the terminal regions of CcmA. The primary amino acid sequence of CcmA from the *H. pylori* G27 genome (GenBank: CP001173.1) was submitted to SWISS-MODEL (https://swissmodel.expasy.org/; *Waterhouse et al., 2018*) where the structure of BacA from *C. crescentus* (*Vasa et al., 2015*) was used to model CcmA. RoseTTAFold was used to predict the structure of CcmA; the same primary amino acid sequence of CcmA was submitted to the Robetta protein prediction service (https://robetta.bakerlab.org/; *Baek et al., 2021*). Models generated in RoseTTAFold were visualized using PyMOL (The PyMOL Molecular Graphics System, version 2.4.2 Schrödinger, LLC).

## Western blotting

Whole-cell extracts of *H. pylori* were prepared by harvesting 1.0 optical density at 600 nm (OD$_{600}$) of log phase liquid culture (0.3–0.7 OD$_{600}$) by centrifugation for 2 min. Then, bacteria were resuspended

to a concentration of 10 $OD_{600}$ per ml with 2× protein sample buffer (0.1 M Tris–HCl, 4% SDS, 0.2% bromophenol blue, 20% glycerol) with or without 5% β-mercaptoethanol (BME). Samples were loaded onto Mini-PROTEAN TGX 4–15% polyacrylamide gels (Bio-Rad) and transferred onto polyvinylidene difluoride (PVDF) membranes using the Trans-Blot TurboTransfer System (Bio-Rad) system. Membranes were either stained with 0.1% Ponceau S solution in 5% acetic acid (Sigma) first or immediately blocked for 2 hr at room temperature in 5% non-fat milk in Tris-buffered saline containing 0.05% Tween-20 (TBS-T), then incubated either overnight at 4°C or at room temperature for 1 hr with primary antibodies, 1:10,000 for α-CcmA (*Blair et al., 2018*), 1:5000 for α-FLAG (M2, Sigma), 1:10,000 for α-Csd1 (*Yang et al., 2019*), 1:20,000 for α-cag3 (*Pinto-Santini and Salama, 2009*), 1:5000 for α-SH3 (*Blair et al., 2018*), or 1:10,000 for α-HaloTag (Promega). Then, it was washed five times with TBS-T over a 30 min period, incubated with secondary antibodies for 1 hr at room temperature at 1:20,000 with appropriate horseradish peroxidase-conjugated α-immunoglobulin antibodies (α-rabbit for CcmA, Csd1, SH3, or Cag3; α-mouse for FLAG and HaloTag, Santa Cruz Biotechnology), and then washed five times with TBS-T over a 30 min period. Proteins were detected using a chemiluminescent substrate according to the manufacturer's instructions (Pierce ECL substrate for standard Western blotting or Immobilon Western HRP substrate [Millipore] for Western blotting following co-immunoprecipitation experiments) and imaged directly with the Bio-Rad gel documentation system. Some membranes were stripped and re-probed with different antibodies, in that case membranes were first rehydrated in TBS-T then incubated in low pH stripping buffer for 20 min while shaking at room temperature (200 mM glycine, 3.5 mM SDS, 1% Tween-20, pH 2.2). After stripping, membranes were washed three times with TBS-T for 5 min each before proceeding to the blocking step.

## Western blot densitometry

Western blot densitometry was performed using Image Lab software version 6.0.1 (Bio-Rad). The lanes on each Western blot were defined, then the adjusted total lane volume was calculated by the software. To calculate relative Csd1 levels, the signal detected by the α-Csd1 antibody for each sample was divided by the signal detected by the α-Cag3 antibody from the same sample on the same membrane. After calculating relative Csd1 levels, they were normalized by dividing the signal by the relative Csd1 level from the SSH51A strain (*rdxA::csd1*).

## Phase-contrast microscopy and quantitative morphology analysis

Phase-contrast microscopy and image analysis was performed as previously described (*Sycuro et al., 2010*). Briefly, cells were grown in liquid culture to mid-log phase (0.3–0.7 $OD_{600}$), centrifuged, and resuspended in 4% paraformaldehyde (PFA) in phosphate-buffered saline (PBS) and mounted on glass slides. Cells were imaged with a Nikon TE 200 microscope equipped with a ×100 oil immersion lens objective and Photometrics CoolSNAP HQ charge-coupled-device camera. Images were thresholded with Fiji (*Schindelin et al., 2012*) and CellTool software (*Pincus and Theriot, 2007*) was used to measure both side curvature and central axis length.

## Protein purification

Purification of 6-His CcmA was performed exactly as described previously (*Taylor et al., 2020*). *E. coli* strains used to recombinantly express CcmA were pKB62A in BL21 DE3, pSS15A in BL21 DE3, pSS16A in BL21 DE3, pSS17A in BL21 DE3, pSS18B in BL21 DE3, pSS19A in BL21 DE3, pKB69H in BL21 DE3, and pKB72D in BL21 DE3.

## Circular dichroism

After purification of 6-his CcmA, samples were dialyzed against 10 mM CAPS pH 11 at a volume of 1:10,000 for 24 hr, then the buffer was changed, and dialysis continued for another 24 hr. After dialysis was complete, samples were normalized to 0.5 mg/ml. CD spectra were collected from 180 to 260 nm while holding the temperature constant with a Jasco J-815 Circular Dichroism Spectropolarimeter.

## Negative stain and TEM

After purification of 6-his CcmA, samples were dialyzed against 25 mM Tris pH 8 for 24 hr. Samples were then normalized to 1 mg/ml and placed on carbon-coated grids after glow-discharging. Samples were fixed in ½ X Karnovsky's fixative and washed with 0.1 M cacodylate buffer and water, then

stained with Nano-W (Nanoprobes) for 60 s, wicked, then stained for another 60 s. Images were acquired on a kV Thermo Fisher Talos 120C LaB$_6$ microscope equipped with a Ceta Camera using Leginon software (*Suloway et al., 2005*). The width of bundles and filaments was measured using Fiji (*Schindelin et al., 2012*).

## Co-immunoprecipitation experiments

Co-immunoprecipitation experiments were performed as previously described in *Blair et al., 2018* and *Yang et al., 2019* with slight modifications. *H. pylori* expressing FLAG tags fused to either Csd5 or Csd7 were grown to mid-log phase in liquid culture and 15 OD$_{600}$ were harvested by centrifugation at 10,000 RPM for 10 min at 4°C. Cells pellets were resuspended in 1.8 ml of cold lysis buffer (20 mM Tris pH 8, 150 mM NaCl, 2% glycerol, 1% Triton X-100, and EDTA-free protease inhibitor [Pierce]). The cells were sonicated at 10% power for 10 s intervals with a microtip (Sonic Dismembrator Model 500, Branson) until visibly cleared, then centrifuged at 20,000 RCF for 30 min at 4°C. The soluble fraction was then incubated with 40 µl beads (α-FLAG M2 agarose beads [Sigma]) or α-VSV-G agarose beads (Sigma) equilibrated in wash buffer (20 mM Tris pH 8, 150 mM NaCl, 2% glycerol, 0.1% Triton X-100) for 90 min at 4°C. The beads were then washed four times in 10 ml cold wash buffer, and samples were centrifuged at 2000 RPM for 2 min between washes. Beads were then boiled in 45 µl 2× sample buffer without BME for 10 min and subjected to Western blotting.

## Labeling with fluorescent WGA and HaloTag ligands

*H. pylori* expressing CcmA-HaloTag were grown to mid-log phase and fixed with 4% PFA for 45 min at room temperature. After fixing, bacteria were permeabilized for 1 hr at room temperature with PBS + 0.05% Triton X-100. Bacteria were then resuspended in PBS and centrifuged onto #1.5 coverslips. Next, coverslips were incubated on a droplet of PBS containing 30 µg/ml af488-conjugated WGA (Invitrogen) and 4 µM JF-549 (Promega) for 30 min at room temperature. Coverslips were washed four times (10 min each) by placing on fresh droplets of PBS + 0.1% Tween-20, then mounted on glass slides with ProLong Diamond Antifade Mountant (Invitrogen).

## 3D structured illumination microscopy

3D SIM was performed on a DeltaVision OMX SR equipped with PCO scientific CMOS cameras, 488 nm and 568 nm lasers, and an Olympus ×60/1.42 U PLAN APO oil objective with oil matched for the sample refractive index. The DeltaVision OMX SR has a resolution of approximately 100 nm in the X and Y directions and 300 nm in Z. 512 × 512 pixel Z-stack images were acquired with 125 nm spacing and 3 µm total thickness. Figures were generated by changing channel colors, adjusting brightness and contrast in Fiji, and brightness and contrast settings were maintained across the entire image (*Schindelin et al., 2012*). Figures were assembled in Adobe Illustrator.

## 3D reconstructions and curvature enrichment analysis

3D reconstructions were generated, and curvature CcmA enrichment analysis was performed exactly as described in *Taylor et al., 2020* using existing software (*Bratton et al., 2019*; *Bratton et al., 2018*). Briefly, 3D SIM Z-stack images were used to generate a 3D triangular meshwork surface with roughly 30 nm precision from the WGA channel. After generating the 3D triangular meshwork, the Gaussian curvature was calculated at every location on the surface of the cell, and we performed curvature enrichment analysis to identify which GC values CcmA signal was associated with where uniform signal = 1. Because the absolute amount of CcmA signal can differ between cells and because the illumination throughout the field is nonuniform, we set the average CcmA signal on each cell as 1. Then we measured each cell's curvature-dependent CcmA signal intensity relative to 1, normalized by the amount of the curvature present on that individual cell surface.

## Analysis of intensity of CcmA-HaloTag signal at cell surface and within cells

A custom MATLAB script was used to compute the relative intensity of CcmA-HaloTag signal at the cell surface and at surfaces generated by shrinking the surface of interest toward the centerline of the cell (negative offsets) or increasing the surface away from the centerline (positive offsets). For each of these shells, or surface offsets, we calculated the average WGA signal and the average CcmA-HaloTag

signal. To account for errors in spatial alignment between the two color channels, before calculating the intensity at each shell, we performed a rigid body alignment procedure to maximize the intensity of the signal contained within the nominal reconstructed surface (0 nm offset). Consistent with the fact that we utilized the WGA channel for shape reconstructions, this alignment algorithm returned offsets of less than one pixel (one pixel = 80 nm) for the WGA channel. The CcmA-HaloTag channel showed more variability in the offsets, sometimes requiring an offset of almost 200 nm along the focal direction, consistent with small, wavelength-dependent variations in focal position. After calculating the average intensity at each subshell, or resized surface, we normalized the highest concentration to 1 before averaging across all the cells in our sample.

## Acknowledgements

This research was supported by the US National Institute of Health (NIH) grants F31 AI152331 (SRS), T32 GM95421 (SRS), and R01 AI136946 (NRS). In addition, the GO-MAP Graduate Opportunity Program Research Assistantship award (GOP Award) (SRS) and the VUMC Discovery Scholars in Health and Medicine Program (BPB). This research was supported by the Genomics and Cellular Imaging and Electron Microscopy shared resources of the Fred Hutch/University of Washington Cancer Consortium (P30 CA015704). The opinions, findings, and conclusions or recommendations expressed in this material contents are solely the responsibility of the authors and do not necessarily represent the official views of the NIH.

We thank Joseph Stembel (University of Washington) for construction of strains JS02 and JS09, Jennifer Taylor (University of Washington) for helpful advice and training, Ethan Garner (Harvard) for helpful discussion, James McDermott for thoughtful discussion and review of this manuscript, David Baker and Justin Decarreau (University of Washington), and the support of the Audacious Project at the Institute for Protein Design for use of the DeltaVision OMX microscope, Roland Strong and Barry Stoddard (Fred Hutchinson Cancer Center) for use of the Circular Dichroism Spectropolarimeter, and Caleigh Azumaya (Fred Hutchinson Cancer Center) for assistance with the Talos microscope.

## Additional information

### Funding

| Funder | Grant reference number | Author |
|---|---|---|
| National Institute of Allergy and Infectious Diseases | F31 AI152331 | Sophie R Sichel |
| National Institute of Allergy and Infectious Diseases | R01 AI136946 | Nina R Salama |
| National Institute of General Medical Sciences | T32 GM95421 | Sophie R Sichel |
| GO-MAP Graduate Opportunity Program Research Assistantship Award | | Sophie R Sichel |
| VUMC Discovery Scholars in Health and Medicine Program | Benjamin Bratton | Benjamin P Bratton |
| National Cancer Institute | P30 CA015704 | Nina R Salama |
| Audacious Project | Institute for Protein Design | Sophie R Sichel |

The funders had no role in study design, data collection and interpretation, or the decision to submit the work for publication.

### Author contributions

Sophie R Sichel, Conceptualization, Data curation, Formal analysis, Funding acquisition, Investigation, Writing – original draft, Writing – review and editing; Benjamin P Bratton, Data curation, Software,

Formal analysis, Funding acquisition, Methodology, Writing – review and editing; Nina R Salama, Conceptualization, Supervision, Funding acquisition, Writing – original draft, Project administration, Writing – review and editing

## Author ORCIDs
Sophie R Sichel http://orcid.org/0000-0001-5079-189X
Benjamin P Bratton http://orcid.org/0000-0003-1128-2560
Nina R Salama http://orcid.org/0000-0003-2762-1424

## Decision letter and Author response
Decision letter https://doi.org/10.7554/eLife.80111.sa1
Author response https://doi.org/10.7554/eLife.80111.sa2

## Additional files

### Supplementary files
• MDAR checklist

### Data availability
Data generated or analysed during this study are included in the manuscript and supporting file; Source Data files have been provided for Figures 2, 5, 6 and 7. Microscopy data are available at BioImage Archive and accession code is S-BIAD462.

The following dataset was generated:

| Author(s) | Year | Dataset title | Dataset URL | Database and Identifier |
|---|---|---|---|---|
| Sichel SR | 2022 | Microscopy data generated in the study "Distinct regions of H. pylori's bactofilin regulate protein interactions to control cell shape" | https://www.ebi.ac.uk/biostudies/BioImages/studies/S-BIAD462 | BioImage Archive, S-BIAD462 |

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
