## [Editor Report]

This study addresses an important unresolved question, the mechanisms governing cell shape in a helical organism. Particularly exciting is the observation that a bactofilin – typically viewed as a cytoskeletal protein – plays a key role in modulating the balance between peptidoglycan synthesis and degradation necessary to generate the helical shape. These findings enhance our understanding of cell wall synthesis in bacteria generally, particularly with regard to the importance of local phenomena for cell shape generation.

---

## [Decision Letter]

**Decision letter after peer review:**

Thank you for submitting your article "Distinct regions of *H. pylori's* bactofilin CcmA regulate protein-protein interactions to control helical cell shape" for consideration by *eLife*. Your article has been reviewed by 3 peer reviewers, one of whom is a member of our Board of Reviewing Editors, and the evaluation has been overseen by Wendy Garrett as the Senior Editor. The following individual involved in the review of your submission has agreed to reveal their identity: Alexandre W Bisson Filho (Reviewer #2).

This manuscript investigators key factors that influence cell shape in a helical organism, Helicobacter pylori, with a specific focus on peptidoglycan and the role of bactofilin proteins as modulators of peptidoglycan synthesis and degradation. Reviewers identified a few key experiments that would strengthen the experimental underpinnings of the conclusions and ways to align the manuscript text with the findings.

Essential revisions:

All three reviewers felt that the study addresses an important set of issues including the mechanisms governing cell shape in a helical organism, the factors modulating the balance between peptidoglycan synthesis and degradation, and the role of a bactofilin family member in both processes. At the same time, reviewers expressed concern that the experiments did not fully substantiate the conclusions and that the text is in need of significant revision. In particular:

1) The authors' model indicates that CcmA is a linchpin regulating the activity of Csd5 and Csd7. However, none of the experiments address the possibility that Csd5 and Csd7 might interact on their own with CcmA serving as an inhibitory factor. All assays were conducted in a CcmA+ strain and/or with CcmA in the reaction. It thus remains possible that Csd5 and Csd7 interact directly in the absence of CcmA, in which case CcmA would have a purely inhibitory role. The authors also overlook the observation that a CcmA N-terminal truncation mutant shares the cell shape characteristics of a camp deletion mutant (curved rod; Yang 2019) rather than that of a csd5 deletion mutant (straight rod). How the current CcmA as a linchpin model fits into this observation is unclear. See the response from Reviewer 2 for experimental suggestions to address this set of concerns.

2) More broadly, as written the manuscript is difficult to follow for the non-expert and some of the more striking results could be highlighted more effectively. For example, the finding that CcmA domains not involved in polymerization play an important role in mediating cell shape is new and has potential implications for the function of other bactofilins. This is an exciting result, and the text should be revised to clarify it as such with revisions to the introduction and conclusion. Additionally, revising the text to ensure that the relevant question/hypothesis is stated up front and delineating how each experiment fits into the big picture in the Results section would be very helpful for readers unfamiliar with previous work.

*Reviewer #1 (Recommendations for the authors):*

In its current form, the paper comes across as focused primarily on characterization rather than hypothesis testing which makes it difficult for those outside the immediate field to appreciate. Clearly stating the question/hypothesis up front and delineating how each experiment fits into the big picture in the Results section would be very helpful for readers unfamiliar with previous work (e.g. Based on the presence of the bactofilin domain, we speculated that CcmA interacted with other members of the complex to coordinate peptidoglycan synthesis and remodeling. To test this we first…). Revising the paper with this in mind would improve both readability and impact.

*Reviewer #2 (Recommendations for the authors):*

I really like the paper and I think there should not be much work to be done to get it ready for publication.

From my point of view, the major contribution of this paper is the suggestion that the extreme Nt segment inhibits recruitment of Csd7, favoring the Csd1-Csd7 interaction, and therefore stabilizing Csd1. This implies that the free bactofilin BD is prone to sequester Csd7 from Csd1. There are 2 experiments that could solidify this view:

Is the independent expression of the first 17aa of CcmA under the ΔNt-ccmA background sufficient to block the binding of Csd7 to CcmA-BD and rescue the Csd1-Csd7 interaction (and therefore Csd1 levels) and WT shape phenotype? Does the 17aa peptide pulldown with CcmA?

If Csd1 is really stabilized by the interaction with Csd7, then a Csd1-Csd7 translational fusion should protect Csd1 even under the ΔNt-ccmA background.

*Reviewer #3 (Recommendations for the authors):*

The authors need to directly address whether their co-IP experiments are the result of direct interactions between their FLAG-tagged proteins and CcmA, or whether the presence of CcmA is incidental and results from the pulldown of a larger complex containing all three proteins. Specifically, they should establish whether the interactions they are observing are actually between i) CcmA and Csd5/7 or ii) Csd5 and Csd7. The exclusion of direct Csd5-Csd7 interaction in the absence of CcmA is essential for the model they propose (Figure 8), and could be achieved via several different approaches:

– They could utilize a BATCH approach to demonstrate direct interactions between CcmA and 1) Csd5 and 2) Csd7 as well as the absence of direct interactions between Csd5-Csd7.

– They could conduct a co-IP for FLAG-tagged Csd5 and a HIS-tagged Csd7 to demonstrate whether these proteins co-precipitate in 1) wild-type H. pylori and 2) a ccmA deletion strain.

– They could conduct their ccmA localization study utilizing N-terminal truncation variants of ccmA (wild type and Halo-tagged) and observe whether they still observe no membrane localization in their csd5-deletion background.

---

## [Author Response]

Essential revisions:All three reviewers felt that the study addresses an important set of issues including the mechanisms governing cell shape in a helical organism, the factors modulating the balance between peptidoglycan synthesis and degradation, and the role of a bactofilin family member in both processes. At the same time, reviewers expressed concern that the experiments did not fully substantiate the conclusions and that the text is in need of significant revision. In particular:1) The authors' model indicates that CcmA is a linchpin regulating the activity of Csd5 and Csd7. However, none of the experiments address the possibility that Csd5 and Csd7 might interact on their own with CcmA serving as an inhibitory factor. All assays were conducted in a CcmA+ strain and/or with CcmA in the reaction. It thus remains possible that Csd5 and Csd7 interact directly in the absence of CcmA, in which case CcmA would have a purely inhibitory role. The authors also overlook the observation that a CcmA N-terminal truncation mutant shares the cell shape characteristics of a camp deletion mutant (curved rod; Yang 2019) rather than that of a csd5 deletion mutant (straight rod). How the current CcmA as a linchpin model fits into this observation is unclear. See the response from Reviewer 2 for experimental suggestions to address this set of concerns.

We appreciate the reviewing team’s suggestion to explore an alternative hypothesis for the data presented where CcmA inhibits interactions between Csd5 and Csd7. It turns out that Kris Blair, the graduate student who initially characterized the Csd5 complex using mass spectrometry (Blair *et al.,* 2018), did include in his now published thesis (Blair, 2018) an IP-mass spectrometry experiment pulling down Csd5-FLAG in a *∆ccmA* strain. He did not detect Csd7. However, Csd7 is a multipass transmembrane protein, and thus may not be detected well by mass spectrometry (Yang *et al.,* 2019). We thus probed our previous and a new Csd7-FLAG pull down experiment using the *∆ccmA* and *∆NT ccmA* strains with our polyclonal Csd5 antibody. We do not detect an interaction between Csd5 and Csd7 (Figure 5C). This further reinforces our conclusion that CcmA participates in two distinct complexes, one with Csd5 that directs CcmA to the cell envelope and a transient complex with Csd7 that in wild-type cells can only be detected by crosslinking and which is enhanced when the N-terminal region is removed from CcmA.

We realize that these results make one aspect of the model presented in the original Figure 8B likely incorrect. In that model we showed Csd7 being recruited by CcmA to the Csd5-MurF-CcmA complex. Our data instead supports a model where Csd7 can interact with Csd1 and Csd2 or it can interact with CcmA in a complex that does not include Csd5. Our immunoprecipitation experiment in *H. pylori* cells indicates that the two CcmA complexes are mutually exclusive. In further support of this model, we followed Reviewer’s 3 suggestion to examine the phenotype of a *∆csd5ccmA∆NT* strain, which displays curved rod morphology (Figure 7 supplement 2A,B) similar to a *∆csd5∆csd1* mutant (curved rod (Sycuro *et al.,* 2012)) and similar to *∆csd5∆csd7* (Figure 7). Though Csd5 is not present to direct CcmA ∆NT to the inner membrane CcmA ∆NT and Csd7 still interact at high levels causing diminished Csd1 protein levels (Figure 7 Supplement 2C) and a curved rod phenotype. To accommodate this new knowledge, we have added the reference to Kris’ thesis, added the Csd5 antibody detection and *∆ccmA* strain the IP experiment in Figure 5C, modified the model presented in Figure 8B, added the phenotype of the *∆csd5ccmA∆NT* strain as a new supplement to Figure 7 (Figure 7 Supplement 2) and addressed these data and the refined model with new text in the discussion.

2) More broadly, as written the manuscript is difficult to follow for the non-expert and some of the more striking results could be highlighted more effectively. For example, the finding that CcmA domains not involved in polymerization play an important role in mediating cell shape is new and has potential implications for the function of other bactofilins. This is an exciting result, and the text should be revised to clarify it as such with revisions to the introduction and conclusion. Additionally, revising the text to ensure that the relevant question/hypothesis is stated up front and delineating how each experiment fits into the big picture in the Results section would be very helpful for readers unfamiliar with previous work.

To address this suggestion, we revised the text to add more description of the rationale and hypothesis for each experiment we performed within the Results section of the manuscript. Additionally, we revised the Discussion section of the manuscript to connect the results, rationale, and hypotheses together more thoroughly.

In addition to addressing these essential revisions needed for publication, we have responded to the individual comments of the three reviewers below. In most cases these suggestions helped us further improve the manuscript. In a few cases we clarify why we either disagree or feel the suggested experiment/line of investigation should be pursued in future work.

Reviewer #1 (Recommendations for the authors):In its current form, the paper comes across as focused primarily on characterization rather than hypothesis testing which makes it difficult for those outside the immediate field to appreciate. Clearly stating the question/hypothesis up front and delineating how each experiment fits into the big picture in the Results section would be very helpful for readers unfamiliar with previous work (e.g. Based on the presence of the bactofilin domain, we speculated that CcmA interacted with other members of the complex to coordinate peptidoglycan synthesis and remodeling. To test this we first…). Revising the paper with this in mind would improve both readability and impact.

As described in the response to essential revision 2, we revised the text to add more description of the rationale and hypothesis for each experiment we performed within the Results section of the manuscript. Additionally, we revised the Discussion section of the manuscript to connect the results, rationale, and hypotheses together more thoroughly.

Reviewer #2 (Recommendations for the authors):I really like the paper and I think there should not be much work to be done to get it ready for publication.From my point of view, the major contribution of this paper is the suggestion that the extreme Nt segment inhibits recruitment of Csd7, favoring the Csd1-Csd7 interaction, and therefore stabilizing Csd1. This implies that the free bactofilin BD is prone to sequester Csd7 from Csd1. There are 2 experiments that could solidify this view:Is the independent expression of the first 17aa of CcmA under the ΔNt-ccmA background sufficient to block the binding of Csd7 to CcmA-BD and rescue the Csd1-Csd7 interaction (and therefore Csd1 levels) and WT shape phenotype? Does the 17aa peptide pulldown with CcmA?

We agree that this is an interesting idea. We attempted to express a copy of just the first 17 amino acids of CcmA at a secondary locus in our *ccmA ∆NT* strain, but the results were inconclusive. The curved rod shape phenotype of the *ccmA ∆NT* strain was not rescued when we attempted to express the N-terminus in trans. However, we were unable to validate whether the first 17 amino acids were expressed in this strain by Western blotting. To fully address this suggestion, much more optimization is needed. It is likely that we would need to use a large protein tag fused to the N-terminus to stabilize this small peptide and resolve whether it is functional in trans.

If Csd1 is really stabilized by the interaction with Csd7, then a Csd1-Csd7 translational fusion should protect Csd1 even under the ΔNt-ccmA background.

While this is an interesting idea, we do not know the topology of Csd7 and need Csd1 in the periplasm to be functional. It’s also not clear how we would interpret a negative result. As this experiment is not essential for publication, we would like to pursue this and similar experiments to build on the results presented in future work.

Reviewer #3 (Recommendations for the authors):The authors need to directly address whether their co-IP experiments are the result of direct interactions between their FLAG-tagged proteins and CcmA, or whether the presence of CcmA is incidental and results from the pulldown of a larger complex containing all three proteins. Specifically, they should establish whether the interactions they are observing are actually between i) CcmA and Csd5/7 or ii) Csd5 and Csd7. The exclusion of direct Csd5-Csd7 interaction in the absence of CcmA is essential for the model they propose (Figure 8), and could be achieved via several different approaches:– They could utilize a BATCH approach to demonstrate direct interactions between CcmA and 1) Csd5 and 2) Csd7 as well as the absence of direct interactions between Csd5-Csd7.– They could conduct a co-IP for FLAG-tagged Csd5 and a HIS-tagged Csd7 to demonstrate whether these proteins co-precipitate in 1) wild-type H. pylori and 2) a ccmA deletion strain.– They could conduct their ccmA localization study utilizing N-terminal truncation variants of ccmA (wild type and Halo-tagged) and observe whether they still observe no membrane localization in their csd5-deletion background.

As described in the response to essential revision 1, we appreciate the reviewing team’s suggestion to explore an alternative hypothesis for the data presented where CcmA inhibits interactions between Csd5 and Csd7. It turns out that Kris Blair, the graduate student who initially characterized the Csd5 complex using mass spectrometry (Blair *et al.*, 2018), did include in his now published thesis (Blair, 2018) an IP-mass spectrometry experiment pulling down Csd5-FLAG in a *∆ccmA* strain. He did not detect Csd7. However, Csd7 is a multipass transmembrane protein, and thus may not be detected well by mass spectrometry (Yang *et al.*, 2019). We thus probed our previous and a new Csd7-FLAG pull down experiment using the *∆ccmA* and *∆NT ccmA* strains with our polyclonal Csd5 antibody. We do not detect an interaction between Csd5 and Csd7 (Figure 5C). This further reinforces our conclusion that CcmA participates in two distinct complexes, one with Csd5 that directs CcmA to the cell envelope and a transient complex with Csd7 that in wild-type cells can only be detected by crosslinking and which is enhanced when the N-terminal region is removed from CcmA.

We realize that these results make one aspect of the model presented in the original Figure 8B likely incorrect. In that model we showed Csd7 being recruited by CcmA to the Csd5-MurF-CcmA complex. Our data instead supports a model where Csd7 can interact with Csd1 and Csd2 or it can interact with CcmA in a complex that does not include Csd5. Our immunoprecipitation experiment in *H. pylori* cells indicates that the two CcmA complexes are mutually exclusive. In further support of this model, we followed Reviewer’s 3 suggestion to examine the phenotype of a *∆csd5ccmA∆NT* strain, which displays curved rod morphology (Figure 7 supplement 2A,B) similar to a *∆csd5∆csd1* mutant (curved rod (Sycuro *et al.,* 2012)) and similar to *∆csd5∆csd7* as shown in Figure 7. Though Csd5 is not present to direct CcmA ∆NT to the inner membrane, this indicates that CcmA ∆NT and Csd7 still interact at high levels causing diminished Csd1 protein levels (Figure 7 Supplement 2C) and a curved rod phenotype. To accommodate this new knowledge, we have added the reference to Kris’ thesis, added the Csd5 antibody detection and *∆ccmA* strain the IP experiment in Figure 5C, modified the model presented in Figure 8B, added the phenotype of the *∆csd5ccmA∆NT* strain as a new supplement to Figure 7 (Figure 7 supplement 2) and addressed these data and the refined model with new text in the discussion.

Line 61-64:A minor point, but the probe used in this study (a modified D-alanine) is thought to incorporate peptidoglycan by D,D- and L,D- transpeptidases. As such, its incorporation is a readout of these processes, rather than 'increased levels of PG synthesis'. In order to more accurately measure new PG synthesis activity, D-ala dipeptide probes would be more accurate, as they are thought to be incorporated during the cytoplasmic stages of PG biosynthesis, specifically by the MurF enzyme (Liechti 2014,2016 and Kuru 2019). - Given this and the fact that MurF appears to be incorporated into the 'helical-cellshape-promoting protein complex' previously defined by the authors, it is curious that the authors did not make use of the dipeptide probes in this (and previous) studies.

In the study the reviewer is referring to (Taylor et al, 2020) both MurNAc-alk and D-Ala-alk were used to visualize new PG incorporation and both probes gave similar results. Similar to D-ala dipeptides probes, we showed that MurNAc-alk is incorporated into the PG sacculus exclusively during de novo PG synthesis. We agree that future studies that examine PG incorporation patterns in various mutational backgrounds would be useful to further dissect the mechanisms of helical cell shape generation in *H. pylori.*

Fig 2:The fact that side curvature is enhanced in the truncation mutant that lacks the authors' defined 'membrane spanning region' of CcmA should be discussed further. A localization study would also seem prudent (particularly as the authors have demonstrated the techniques required later in the manuscript) demonstrating that MM is necessary and sufficient for membrane association (GFP-fusion study) while amino acids 13-17 are not; alternatively, the absence of membrane association would effectively demonstrate that the N-terminal domain is not involved in membrane association.

In Figure 7, we showed that Csd5 is required for CcmA to localize to the inner membrane. In Figure 5 we show that the membrane binding motif is not required for CcmA interaction with Csd5. Based on these two results we conclude that the putative membrane binding motif proposed by another group (amino acids 3-12) (Deng et al., 2019) does not mediate membrane binding for CcmA in H. pylori. These points are now more clearly stated in the results and discussion.

The Cag3 band looks different in CT and BD lanes. Did the authors use more protein than they did in the other wells?

While we tried to do our best to evenly load all of the lanes with H. pylori lysates, it does look like we loaded slightly less lysate in these two lanes in this particular blot. However, the slightly lower loading does not detract from the overall conclusion of the western blot – all truncation mutant versions of CcmA are expressed in H. pylori cells. As stated in the figure legend, the blot shown is representative of two independent experiments.

Fig 3:The addition of the 2-point mutants enables a good comparison with the truncation mutants. For continuity, it would be helpful if they were also used in the experiments conducted in Figure 2 (protein expression, cell shape, and side curvature)

Assays to identify if CcmA I55A and L110S point mutants impact protein expression, cell shape, and side curvature were all performed and described in a previous publication (Taylor et al., 2020).

As a matter of thoroughness, the 95C condition should be shown for both the L110S and I55A variants. Perhaps panels A and B could be redone/rearranged with a 20C (A) and 95c (B) condition for each CcmA protein.

We did collect CD spectra on I55A and L110S at both 95 ºC and 20 ºC, but we did not include the datasets from 95 ºC, because the data did not contribute any new insight. We are including the data here (below). Both I55A and L110S look the same at 95 ºC as they do at 20 ºC because the proteins are unfolded.

Lines 211-230; Fig 4:This section is very descriptive and largely highlights phenotypes described by the authors in their previous work. While they do quantitate bundle width, the rest of their observations involve applying largely qualitative assessments of these structures to their newly made truncation mutants. It would enhance the section greatly (and build substantially on their previous efforts) if the authors made an attempt to quantify some of the other descriptors they mention: filament vs bundle width, "filaments and bundles... more homogenous, straighter, and thinner than WT ...", "...appear to only contain 2-4 filaments each", etc.

We agree with the reviewer that our characterization of the TEM data is qualitative, not quantitative. Given the limitations of TEM sample preparation and imaging and the lack of a specific recommendation by the reviewer for a software package or approach to perform more quantitative measurements, we feel this level of analysis is appropriate for the data.

Readers unfamiliar with the author's previous work may not appreciate the lattice structures referred to by the authors from the images provided. Recommend enhancing the enlarged images and/or adjusting brightness/contrast to better visualize the lattice structures Response: We updated the figure with micrographs that have higher contrast to help the reader see the filaments, bundles, and lattices.

We updated the figure with micrographs that have higher contrast to help the reader see the filaments, bundles, and lattices.

Granted the limitations inherent to TEM sample preparation/imaging, it would be informative if the authors shared the frequency of lattice structure formation in their positive lattice samples as well as the largest distances between filaments in which lattice structures are observed

In our previous work ((Taylor et al., 2020), Figure 8 Supplement 2) we characterized the lattices formed by WT CcmA by Fourier analysis. Given the limitations noted of TEM sample preparation/imaging, we do not feel confident in measurements of how frequently lattice structures are formed. Further studies of polymerization in vitro will require development of new methods beyond the scope of this manuscript.

Fig 5:- As stated previously, it is not entirely intuitive whether the association observed by the authors between Csd7-FLAG and their NT and BD truncation mutants (in Panel B) is the result of direct or indirect interaction.- The authors conclude that "the N-terminal region of CcmA inhibits the interaction with Csd7", but it is telling that they see this inhibition in both their MM and 13-17 mutants.- Given that the authors have not precisely characterized the membrane-spanning portion of the N-terminal region of their protein (their proposed motif is amino acids 3- 12; lines 131-132), how do they know that the interaction is not simply blocked by the bactofilin being associated with the membrane?- This raises the obvious follow-up question: when during H. pylori's growth/division/membrane biogenesis processes would the bactofilin no longer be in contact with the inner membrane (thus enabling Csd7 to interact with CcmA)?- A more robust demonstration of direct protein-protein interactions between Csd7, Csd5, and CcmA is warranted- Alternatively, with the addition of a separate tag (example, Csd5/7-HIS) the authors could demonstrate whether they see Csd7 in their Csd5 pull downs (and vice versa) in their wild type and truncated ccmA mutant strains (as well as in a ccmA deletion mutant)

As stated in the response to essential revision 1, we appreciate the reviewing team’s suggestion to explore an alternative hypothesis for the data presented where CcmA inhibits interactions between Csd5 and Csd7. It turns out that Kris Blair, the graduate student who initially characterized the Csd5 complex using mass spectrometry (Blair et al., 2018), did include in his now published thesis (Blair, 2018) an IP-mass spectrometry experiment pulling down Csd5-FLAG in a ΔccmA strain. He did not detect Csd7. However, Csd7 is a multipass transmembrane protein, and thus may not be detected well by mass spectrometry (Yang et al., 2019). We thus probed our previous and a new Csd7-FLAG pull down experiment using the ΔccmA and ΔNT ccmA strains with our polyclonal Csd5 antibody. We do not detect an interaction between Csd5 and Csd7 (Figure 5C). This further reinforces our conclusion that CcmA participates in two distinct complexes, one with Csd5 that directs CcmA to the cell envelope and a transient complex with Csd7 that in wild-type cells can only be detected by crosslinking and which is enhanced when the N-terminal region is removed from CcmA.

Fig. 7; Lines 352-354:A major caveat to this experimentation is that functional studies will be of chimeric filaments likely composed of both wild-type and Halo-tagged CcmA. It is unclear whether polymerization kinetics of CcmA and CcmA-HALO are similar; Have the authors attempted to demonstrate that these chimeric filaments incorporating HALOtagged proteins exhibit altered filamentation/lattice formation (confirmed by TEM)?

We have not looked at whether CcmA-HaloTag forms lattices, all experiments involving CcmA-HaloTag have been performed in H. pylori cells, we have not recombinantly expressed and purified CcmA-HaloTag from E. coli. However, we believe that CcmA-HaloTag can polymerize because cells expressing a single copy of CcmA-HaloTag (without WT CcmA co-expressed) are still helical and more have more curvature than ∆ccmA cells, which indicates that CcmA-HaloTag is functional (Figure 7 – supplement 1).

It is notable that no analysis was conducted on the number or average length/size of the puncta and filaments formed and whether they differ significantly between mutant strains

Unfortunately, at present we do not know the relationship between the filaments, bundles and lattices observed by TEM and the puncta visualized by 3D SIM. Correlative light-electron microscopy (CLEM) could possibly get at the issue and certainly would be an interesting future direction of this project.

Additionally, it is still an open question whether CcmA is interacting with Csd5 and inhibiting interaction with Csd7 in a polymerized form or a monomeric form and whether subpopulations of polymerized and monomeric CcmA are present in H. pylori cells. We added to our discussion section to discuss this open question at line 630.

For panel C, it would be helpful to include the Gaussian curvature characteristics of the ccmA deletion mutant as well, so that each strain can be compared against the others.

In panel C, a reconstruction of a ∆ccmA cell was added.

Fig. 8:The model doesn't show the membrane-spanning component of CcmA's NT region; do the authors believe that the domain is not membrane-associated?

In Figure 7 we identified that Csd5 is required for CcmA to localize to the inner membrane, thus CcmA cannot localize to the inner membrane on its own. We do not think the putative membrane binding motif identified by Deng and colleagues (Deng et al., 2019) functions as a membrane binding motif in CcmA. We have clarified this point in the text.

The model illustrates what the authors believe is happening in their mutant strains, but does convincingly explain why the Csd7-CcmA interaction might be important in the wild-type condition.

We added further detail in our discussion section at line 607 to clarify our hypothesis for how the Csd7-CcmA interaction functions in WT cells.

To be relevant, the authors would need some evidence that alterations occur to/at the N-terminal region of CcmA, which enable the Csd7 interaction (and subsequent destabilization of Csd2/Csd1) that they see in their truncation mutant; otherwise, the simple absence of CcmA from a particular area of the cell wall would make more sense

We agree with the reviewer that future biochemical and/or structural work to elucidate how the N-terminus influences CcmA structure, polymerization states in vivo and Csd7 interactions will be important. This will be a focus of future work.

The model also does not incorporate the finding that the NT regions are essential for lattice formation

As stated in the response to the previous point, we agree with the reviewer that future biochemical and/or structural work to elucidate how the N terminus influences CcmA structure and polymerization states in vivo should be a focus of future work. Although we do see that purified CcmA forms lattices in vitro, our experimental evidence suggests that lattice formation is not essential for CcmA’s function in vivo. For example, in Figure 4 we showed that purified CcmA ∆CT does not form lattices in vitro, but Figure 2 shows that H. pylori cells that express ccmA ∆CT are still helical. Together these results suggest that lattice formation in vitro is not required for CcmA to pattern helical cell shape in vivo. Given these results we do not feel comfortable including lattice formation in our current model of CcmA function.

For Panel C, what do the authors predict would occur if their NT truncated ccmA is present in the csd5 mutant background? The answer to this hypothetical (demonstrated experimentally) would provide strong proof of concept for the entire model.

We hypothesize that there could be two possibilities of how CcmA ∆NT would function in a ∆csd5ccmA∆NT strain. (1) Without Csd5, CcmA ∆NT would not be able to localize to the inner membrane and be close enough to interact with Csd7, thus CcmA ∆NT and Csd7 would not interact at high levels. In this case Csd1 should be stable and the shape phenotype would resemble ∆csd5 (straight rod) (2) Even though Csd5 is not present to direct CcmA ∆NT to the inner membrane, interactions between CcmA ∆NT and Csd7 are strong enough to direct CcmA to the inner membrane, thus CcmA ∆NT and Csd7 would still interact at high levels. In this case we would expect diminished Csd1 protein levels and a phenotype similar to a ∆csd5Δcsd1 (curved rod (Sycuro et al, 2012)) and similar to ∆csd5Δcsd7 shown in Figure 7. We have now constructed a ∆csd5ccmA∆NT strain and find it has a curved rod phenotype and lower levels of Csd1 supporting the second possibility, further reinforcing our revised model of separate Csd5 and Csd7 complexes. This new data has been added as Figure 7 Supplement 2 and reinforces our revision to the model presented in Figure 8B.

Line 863-870:It would be helpful for the reader to note the resolution limits of the instrument in the x, y, and Z planes (in each imaging channel).

Resolution is approximately 100 nm in the X and Y directions and 300 nm in Z. This information was added to the text.

Line 896:Please denote pixel scale equivalence (preferably in nm).

Pixels in the X and Y direction are 0.08 µm (80 nm) and 0.125 µm (125 nm) in the Z direction. This was added to the text.

References

Blair K, 2018. Exploring Mechanisms of Cell Shape Control in *Helicobacter pylori*, Molecular and Cellular Biology. University of Washington, Seattle, WA.

Blair KM, Mears KS, Taylor JA, Fero J, Jones LA, Gafken PR, Whitney JC, Salama NR (2018) The Helicobacter pylori cell shape promoting protein Csd5 interacts with the cell wall, MurF, and the bacterial cytoskeleton. *Mol Microbiol* 110: 114-127

Deng X, Gonzalez Llamazares A, Wagstaff JM, Hale VL, Cannone G, McLaughlin SH, Kureisaite-Ciziene D, Lowe J (2019) The structure of bactofilin filaments reveals their mode of membrane binding and lack of polarity. *Nat Microbiol* 4: 2357-2368

Kuhn J, Briegel A, Morschel E, Kahnt J, Leser K, Wick S, Jensen GJ, Thanbichler M (2010) Bactofilins, a ubiquitous class of cytoskeletal proteins mediating polar localization of a cell wall synthase in *Caulobacter crescentus*. *EMBO J* 29: 327-339

Sycuro LK, Wyckoff TJ, Biboy J, Born P, Pincus Z, Vollmer W, Salama NR (2012) Multiple peptidoglycan modification networks modulate Helicobacter pylori's cell shape, motility, and colonization potential. *PLoS Pathog* 8: e1002603

Taylor JA, Bratton BP, Sichel SR, Blair KM, Jacobs HM, DeMeester KE, Kuru E, Gray J, Biboy J, VanNieuwenhze MS *et al.* (2020) Distinct cytoskeletal proteins define zones of enhanced cell wall synthesis in Helicobacter pylori. *ELife* 9

Vasa S, Lin L, Shi C, Habenstein B, Riedel D, Kuhn J, Thanbichler M, Lange A (2015) β-Helical architecture of cytoskeletal bactofilin filaments revealed by solid-state NMR. *Proc Natl Acad Sci U S A* 112: E127-136

Yang DC, Blair KM, Taylor JA, Petersen TW, Sessler T, Tull CM, Leverich CK, Collar AL, Wyckoff TJ, Biboy J *et al.* (2019) A Genome-Wide Helicobacter pylori Morphology Screen Uncovers a Membrane-Spanning Helical Cell Shape Complex. *Journal of Bacteriology* 201: e00724-00718